# Tectal-derived interneurons contribute to phasic and tonic inhibition in the visual thalamus

Polona Jager[1,*], Zhiwen Ye[2,*], Xiao Yu[2], Laskaro Zagoraiou[3], Hong-Ting Prekop[1], Juha Partanen[4], Thomas M. Jessell[5], William Wisden[2,6], Stephen G. Brickley[2,6] & Alessio Delogu[1]

The release of GABA from local interneurons in the dorsal lateral geniculate nucleus (dLGN-INs) provides inhibitory control during visual processing within the thalamus. It is commonly assumed that this important class of interneurons originates from within the thalamic complex, but we now show that during early postnatal development *Sox14/Otx2*-expressing precursor cells migrate from the dorsal midbrain to generate dLGN-INs. The unexpected extra-diencephalic origin of dLGN-INs sets them apart from GABAergic neurons of the reticular thalamic nucleus. Using optogenetics we show that at increased firing rates tectal-derived dLGN-INs generate a powerful form of tonic inhibition that regulates the gain of thalamic relay neurons through recruitment of extrasynaptic high-affinity GABA$_A$ receptors. Therefore, by revising the conventional view of thalamic interneuron ontogeny we demonstrate how a previously unappreciated mesencephalic population controls thalamic relay neuron excitability.

[1] Department of Basic and Clinical Neuroscience, Institute of Psychiatry, Psychology and Neuroscience, King's College London, 125 Coldharbour Lane, London SE5 9NU, UK. [2] Department of Life Sciences, Imperial College London, London SW7 2AZ, UK. [3] Biomedical Research Foundation Academy of Athens, 4 Soranou Ephessiou Street, 115 27 Athens, Greece. [4] Department of Biosciences, University of Helsinki, PO Box 56, 00014 Helsinki, Finland. [5] Howard Hughes Medical Institute, Columbia University, 701 West 168th Street, HHSC Room 1013, New York, New York 10032, USA. [6] Centre for Neurotechnology, Imperial College London, London SW7 2AZ, UK. * These authors contributed equally to this work. Correspondence and requests for materials should be addressed to S.G.B. (email: s.brickley@imperial.ac.uk) or to A.D. (email: alessio.delogu@kcl.ac.uk).

GABAergic neurons play key roles during development, assembly and refinement of neuronal circuits[1,2] before helping to shape patterns of neuronal activity in the mature brain. Whilst the development[3–5] and functional diversity[6–8] of GABAergic neurons have been well described in the telencephalon, less is known about the ontogeny of thalamic inhibition. Within the thalamus, GABAergic drive depends on two functionally distinct neuronal populations: intrinsic interneurons, and neurons of the reticular thalamic nucleus (RTN)[9,10]. The origin of local thalamic interneurons in particular has remained elusive. According to the prosomeric model[11,12], thalamic progenitors are specified within the second (also known as dorsal thalamus) and third (also known as ventral thalamus or prethalamus) diencephalic prosomere (p2 and p3). Inhibitory progenitors, including those of the RTN, are found in p3; whilst excitatory neurogenesis takes place in p2 (refs 13,14). A GABAergic rostral p2 subdomain has also been identified (pTh-R)[15] making both p3 and pTh-R possible sources of origin for thalamic interneurons.

Mapping of expression domains of key transcriptional regulators during embryonic brain development has proven instrumental for the identification of defined cell lineages. GABAergic progenitors in p3, including those of the RTN, express *Ascl1* and progress through a differentiation programme that depends on *Dlx1*, *Dlx2* and *Arx* (refs 11,16–21). GABAergic neurogenesis in pTh-R requires interaction between the pan-GABAergic proneural transcription factor Ascl1 and the GABAergic subtype-specific transcription factor Helt[16]. Ascl1-Helt dimerization leads to sequential activation of other transcription factor genes, including *Gata2*, *Tal1*, *Tal2*, *Gata3* and *Sox14* (refs 16–19,22–24), all of which are also found in the first diencephalic prosomere (p1, also known as the pretectum), but not in p3. Furthermore, reciprocal repression between *Dlx1/2* and *Gata2/3* suggests that alternative GABA fates are acquired in p3 and pTh-R (refs 17–19). A recent report described an *Otx2*-positive and *Sox14*-negative GABAergic lineage in p3 that contributes to local thalamic interneurons[25], providing experimental support to the prevailing hypothesis that dLGN interneurons have a prethalamic origin[26,27]. In contrast, in this study we report that p3, as well as pTh-R and p1, are unlikely sources of thalamic interneurons and propose an alternative model whereby an incoming tectal population seeds the thalamus with inhibitory interneurons in a process requiring *Gata2* and *Sox14*.

Although interneurons are ubiquitous in the thalamus of carnivorans and primates, they are largely confined to the dorsal lateral geniculate nucleus (dLGN) in rodents[28]. Retinal input to the dLGN directly excites local GABAergic interneurons (dLGN-INs)[29–31] providing visually driven feed-forward inhibition of dLGN thalamic relay neurons[32]. The F2 terminals formed by dLGN-INs also display atypical wiring, releasing GABA at dendro-dendritic sites[33] found within triadic structures known as glomeruli[34]. In addition, F1 terminals are more classical axo-dendrtic synaptic arrangements. The recruitment of GABA release at the more specialized triadic structures has been shown to result in the activation of both phasic and tonic forms of inhibition in the dLGN[35]. In contrast, RTN neurons introduce feedback inhibition within the thalamo-cortical pathway, contributing to increased spatial attention[36] and promoting synchronous activity in the thalamo-cortical axis during sleep[37].

With the aim of characterizing the inhibitory properties of these dLGN-INs, we took advantage of their genetic identity and generated a *Sox14^{cre}* knock-in mouse line to drive expression of the light-gated ion channel Channelrhodopsin 2 (ChR2). Using optogenetics we demonstrate that GABA released from *Sox14^+* dLGN-INs generates a frequency-dependent form of tonic inhibition, which depends on the activation of extrasynaptic GABA$_A$ receptors. Therefore, by exploiting the unappreciated ontogeny of dLGN-INs we have developed a novel strategy for controlling visual processing in the thalamus.

## Results

**Transcription factor Sox14 is a genetic marker for dLGN-INs.** We and others have reported that the *Sox14* gene is often associated with GABAergic neurons in subcortical brain regions[15,17,19,22,23], but this association was thought not to apply for local thalamic interneurons[25]. The observation of scattered *Gfp^+* cells within thalamic relay nuclei of *Sox14^{Gfp/+}* mice was therefore, unexpected. To investigate further the identity of these previously unreported *Sox14^+* neurons, we first confirmed their presence within the boundaries of the interneuron-rich dLGN visualized by anterograde labelling of the optic tract (Fig. 1a and Supplementary Movie 1) and subsequently show that they contain the neurotransmitter GABA (Fig. 1b).

To functionally classify the *Sox14^+* neurons, whole-cell recordings were made from *Gfp^+* and *Gfp^−* dLGN cells in acute brain sections prepared from 3- to 4-week-old *Sox14^{Gfp/+}* mice (Fig. 1c). Imaging of neurobiotin fills highlighted the distinct morphology of the *Gfp^+* neurons contrasting with the highly branched, larger dendritic coverage of the *Gfp^−* neurons (Fig. 1d). The characteristic interneuron-like morphology of the *Gfp^+* neurons resulted in a smaller membrane capacitance of $59.9 \pm 5.5$ pF ($n = 43$), compared with $102.2 \pm 7.4$ pF ($n = 32$) for the *Gfp^−* relay neurons ($P < 0.005$, *t*-test; Fig. 1e). The resting input resistance of *Gfp^+* neurons ($1.1 \pm 0.1$ GΩ) was higher ($P < 0.005$, *t*-test) than the *Gfp^−* fraction ($0.4 \pm 0.06$ GΩ), and the current–voltage relationship was much steeper (Fig. 1e). Analysis of the frequency of action potentials (APs) indicates lower maximum firing rates of around 20 Hz for the *Gfp^+* neurons compared with *Gfp^−* neurons. In summary, the membrane properties of the *Gfp^+* fraction and neurotransmitter expression are compatible with the hypothesis that *Sox14^+* neurons are dLGN-INs, whilst all tested *Gfp^−* cells are relay neurons[38,39]. However, a more direct characterization of dLGN-INs was provided by simultaneous paired recordings between the *Gfp^+* and *Gfp^−* neurons. In 9 out of 12 paired recordings increasing the AP firing of the *Gfp^+* putative dLGN-IN resulted in a clear increase in the tonic conductance recorded from the *Gfp^−* thalamic relay neuron (Fig. 1f). The ability of a single dLGN-IN to alter the tonic conductance in a nearby thalamic relay neuron was confirmed by linear regression analysis (Fig. 1g) with $R^2$ values ranging from 0.3 to 0.9 (analysis of variance, $P < 0.005$). The relationship between the firing rate of a single dLGN-IN and the increase in the tonic conductance recorded in the adjacent thalamic relay neurons was on average $31 \pm 17$ pS Hz$^{-1}$ ($n = 12$). In contrast, the low level of direct synaptic connectivity between individual neurons meant that we did not observe a robust relationship between the rate of AP firing in a dLGN-IN and the rate of inhibitory postsynaptic currents (IPSCs) recorded in adjacent thalamic relay neurons (Fig. 1g).

**Sox14 is required for dLGN-IN development.** *Sox14* is required for tangential migration of pTh-R GABAergic precursors between embryonic day (E) 14.5 and 16.5 (ref. 17), and its ablation results in *Sox14^{Gfp/Gfp}* (*Sox14* knockout) embryos having a much-reduced number of *Gfp^+* neurons in the ventral LGN (vLGN) at the time of birth[17]. Because dLGN-INs are known to migrate into the nucleus postnatally[25,26], we extended our previous analysis[17] of the *Sox14* knockout thalamus to the postnatal brain (Fig. 2a). We found that the dLGN of P21 *Sox14^{Gfp/Gfp}* mice was largely devoid of *Gfp^+* neurons with an average of $9.3 \pm 1.0$ *Gfp^+* neurons in each 100 μm-thick section

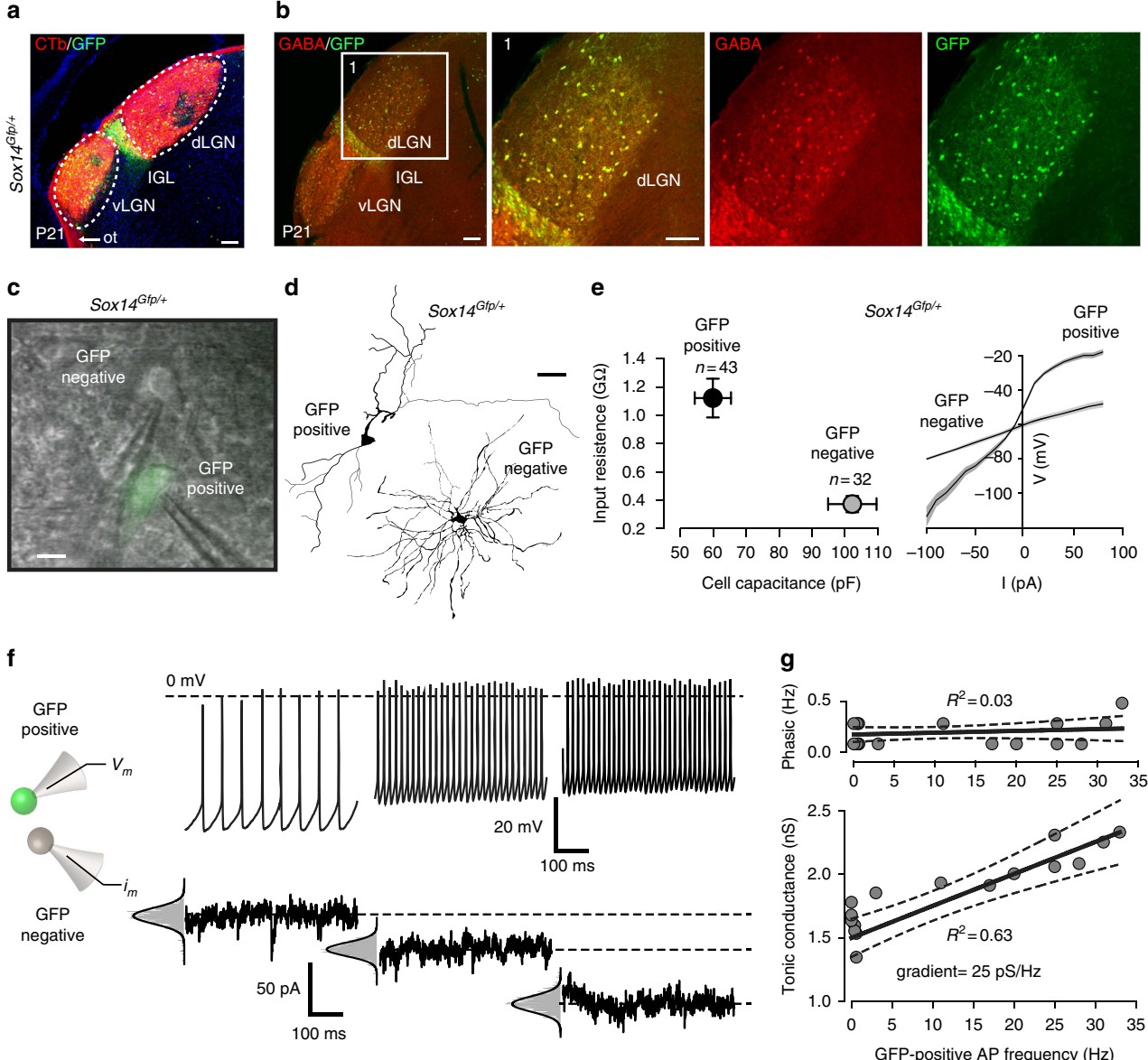

**Figure 1 | Sox14 is a marker of interneurons in the dLGN.** (**a**) Alexa-594-conjugated CTb was used to label the retinal projection to the contralateral LGN. This low-magnification image compares retinal input with the Gfp distribution in the Sox14Gfp/ + mouse brain at P21 3 days after injection. Scale bar, 100 μm (see also Supplementary Movie 1). (**b**) A panel of images showing immunohistochemical detection of GABA-containing neurons in the LGN compared with the distribution of Gfp + neurons in the Sox14Gfp/ + mouse brain at P21. Scale bar, 100 μm. (**c**) Live-cell imaging data of Gfp + and Gfp − neurons in an acute slice prepared from a Sox14Gfp/ + mouse at P27. Patch-clamp electrodes are shown during simultaneous whole-cell recording from dLGN neurons. Scale bar, 30 μm. (**d**) Reconstructions of Gfp + and Gfp − neurons following electrophysiological recording in acute slice preparations of the Sox14Gfp/ + mouse. Scale bar, 50 μm. (**e**) Scatter plot of data comparing cell capacitance and input resistance in Gfp + and Gfp − neurons recorded from the dLGN of Sox14Gfp/ + mice aged between P20 and 27. Mean values are shown with error bars depicting s.e.m. Averaged current–voltage curves for these Gfp + and Gfp − neurons is also shown. The solid lines are the average values with grey areas illustrating s.e.m. (**f**) Data from a simultaneous recording between a Gfp + and Gfp − neuron in an acute slice preparation of the Sox14Gfp/ + mouse. The voltage traces on the top were made during a whole-cell recording from a single Gfp + neuron illustrating how the action potential frequency increases at more depolarized potentials. The current traces are taken from voltage-clamp recordings made simultaneously from a neighbouring Gfp − neuron indicating the increase in tonic conductance as the presynaptic neuron increases firing rate. (**g**) Plot of the average relationship between the AP frequency of a Gfp + neuron and the tonic and phasic conductance changes recorded simultaneously from an adjacent Gfp − neuron. The solid lines illustrate the result of linear regression analysis with the dashed lines demonstrating 95% confidence limits of this fit.

of the dLGN (33 sections covering the entire nucleus, 3 mice) compared with 118.8 ± 23.6 in Sox14Gfp/ + control mice (35 sections covering the entire nucleus, 3 mice). The cell counts (Fig. 2e) indicate that Sox14 is required for the differentiation of this inhibitory subtype. Further supporting the hypothesis that Sox14+ neurons in the dLGN are the only resident inhibitory cell type, the GABA neurotransmitter and the

transcript Gad1 were virtually undetectable in the dLGN of Sox14 knockout mice (Fig. 2a,b), but still present in the nearby RTN and pretectum (Fig. 2b). No increase in apoptosis was detected in Gfp-expressing regions within the LGN region between P0 and P3, suggesting that a migratory failure rather than subsequent apoptosis, leads to the observed lack of dLGN-INs in the Sox14 knockout thalamus (Fig. 2c).

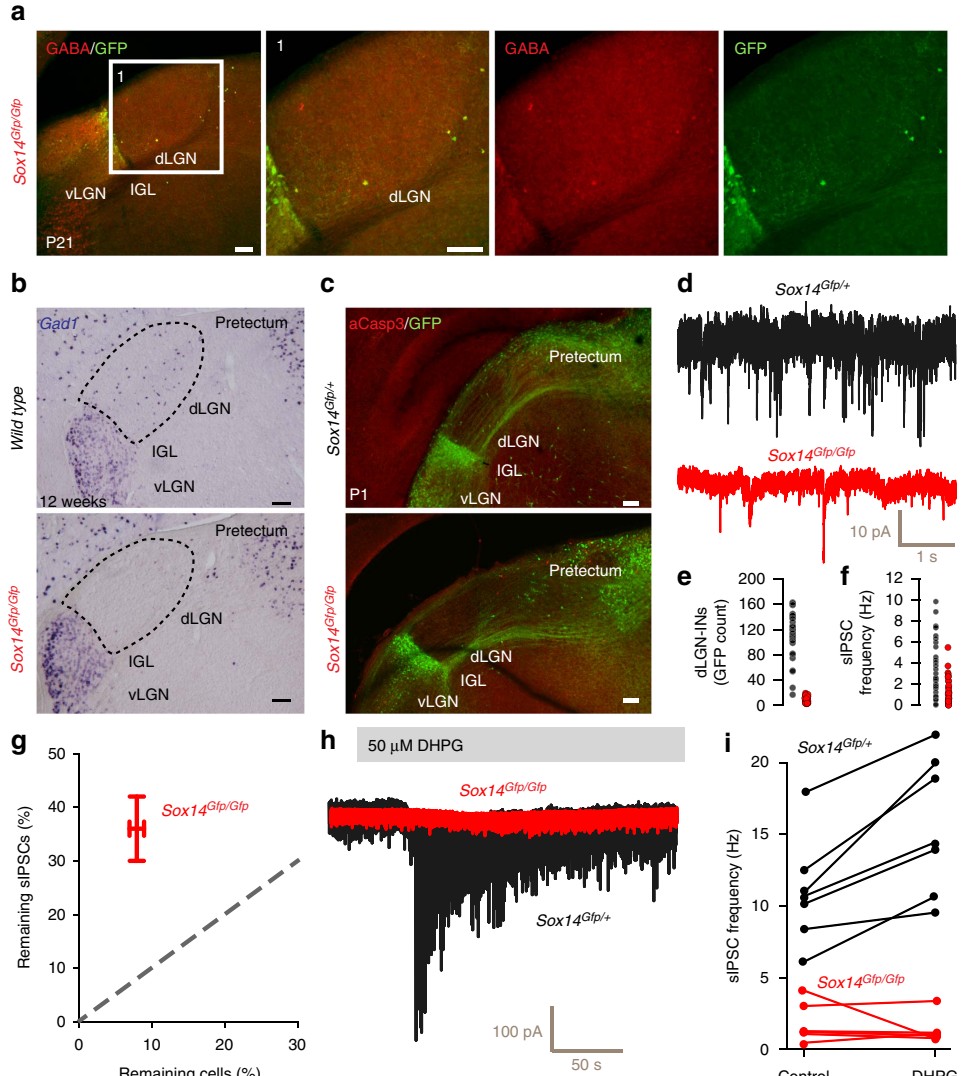

**Figure 2 | *Sox14*$^+$ cells provide GABAergic drive to thalamic relay neurons.** (**a**) Panel of images taken from a knockout *Sox14*$^{Gfp/Gfp}$ mouse illustrating the large reduction in GABA-positive *Sox14* neurons in the dLGN. Scale bar, 100 μm. (**b**) Two ISH images comparing the distribution of *Gad1* mRNA (GAD67 enzyme) in a wild-type (*Sox14*$^{+/+}$) and a *Sox14* knockout (*Sox14*$^{Gfp/Gfp}$) mouse. Note the absence of *Gad1* in the dLGN of the *Sox14* knockout. Scale bar, 100 μm. (**c**) Images demonstrating similar low levels of immunolabelling for aCasp3 in the dLGN of *Sox14*$^{Gfp/+}$ wild-type and *Sox14*$^{Gfp/Gfp}$ knockout brains. Scale bar, 100 μm. (**d**) Example current traces during whole-cell recordings from a relay neuron taken from a *Sox14*$^{Gfp/+}$ wild-type or a *Sox14*$^{Gfp/Gfp}$ knockout mouse. (**e**) Scatter plots for the number of GFP-positive neurons identified in brain slices prepared from either wild-type *Sox14*$^{Gfp/+}$ (black circles) or knockout *Sox14*$^{Gfp/Gfp}$ mice. (**f**) Scatter plots for the spontaneous IPSC (sIPSC) frequency measured in dLGN-INs prepared from either wild-type *Sox14*$^{Gfp/+}$ (black circles) or knockout *Sox14*$^{Gfp/Gfp}$ (red circles) mice. (**g**) A scatter plot comparing the proportion of remaining GFP-positive cells versus the average frequency of sIPSCs recorded from relay neurons in knockout *Sox14*$^{Gfp/Gfp}$ mice (mean ± s.e.m.). (**h**) Two superimposed current traces used to compare the impact of 50 μM DHPG on tonic and phasic inhibition recorded from a wild-type *Sox14*$^{Gfp/+}$ relay neuron (black trace) and a knockout *Sox14*$^{Gfp/Gfp}$ dLGN relay neuron. Note the clear absence of any response in the knockout *Sox14*$^{Gfp/Gfp}$ dLGN relay neuron. (**i**) Scatter plot of the DHPG-induced change in sIPSC frequency for all relay neurons recorded from wild-type *Sox14*$^{Gfp/+}$ (black circles) and knockout *Sox14*$^{Gfp/Gfp}$ mice (red circles). The solid lines link measurements made from individual relays neurons before and after application of DHPG.

**Residual GABAergic drive following removal of dLGN-INs.** The loss of dLGN-INs from the *Sox14*$^{Gfp/Gfp}$ mice did not result in complete removal of spontaneous IPSCs (sIPSCs) from dLGN relay neurons, most likely due to the maintenance of GABA release from RTN terminals. Nonetheless, sIPSC frequency was reduced ($P < 0.005$, unpaired *t*-test) from $3.6 \pm 0.5$ Hz ($n = 31$ cells, 11 mice) in *Sox14*$^{Gfp/+}$ mice to $1.3 \pm 0.2$ Hz ($n = 36$ cells, 6 mice) in *Sox14*$^{Gfp/Gfp}$ mice (Fig. 2d). Therefore, *Sox14*$^+$ dLGN-INs make a significant contribution to GABAergic drive onto thalamic relay neurons (Fig. 2f), and our data could suggest that RTN terminals provide 30% of the spontaneous GABA release

onto thalamic relay neurons, at least in the acute slice preparation (Fig. 2g).

To confirm that the remaining sIPSCs observed in the thalamic relay neurons of *Sox14*$^{Gfp/Gfp}$ mice originated from RTN terminals we took advantage of the fact that the F2 terminals formed by dLGN-INs are confined within the glomerular arrangement of the thalamic triad. The metabotropic glutamate receptor (mGluR) agonist (S)-3,5-dihydroxyphenylglycine (DHPG) is known to increase the rate of vesicular GABA release from dLGN-IN terminals[35,40], and 50 μM DHPG significantly increased ($P < 0.005$, paired *t*-test) the IPSC frequency from

10.9 ± 1.4 to 15.6 ± 1.8 Hz ($n = 7$) in $Sox14^{Gfp/+}$ mice. However, in $Sox14^{Gfp/Gfp}$ mice the remaining GABA release was not affected and sIPSC frequency was 1.8 ± 0.6 Hz ($n = 6$) in control conditions compared with 1.4 ± 0.4 Hz in the presence of DHPG (Fig. 2h,i) indicating a disturbance in the mGluR-mediated activation of GABA release in $Sox14^{Gfp/Gfp}$ mice.

**dLGN-INs generate tonic inhibition in thalamic relay neurons.** Having demonstrated the validity of the *Sox14* marker for identifying dLGN-INs, we then generated a novel *Sox14^cre* knock-in mouse line to enable selective optogenetic activation of dLGN-INs. *Cre*-dependent *AAV-flex-ChR2-eYFP* was injected bilaterally into the LGN of $Sox14^{cre/+}$ mice at P16 (Fig. 3a). After 3 weeks, *Chr2-eYFP*-expressing cell bodies were densely concentrated throughout the intergeniculate leaflet (IGL) and were sparsely distributed throughout the dLGN with clear *Chr2-eYFP* in axonal terminals (Fig. 3a). To demonstrate the functional expression of ChR2 whole-cell recordings were made from dLGN-INs (Fig. 3b). In voltage-clamp configuration, 1 ms duration light-emitting diode (LED) pulses resulted in transient changes in the ChR2-mediated conductance (Fig. 3b grey trace). In current-clamp configuration, these photocurrents elicited robust APs (Fig. 3b, black trace) at LED stimulation rates of up to 30 Hz (Fig. 3b, blue trace). In a total of eight cells, three were clearly able to follow LED stimulation rates up to 30 Hz, but two cells could only respond up to 20 Hz, and three cells could only generate APs up to 10 Hz (Fig. 3c). However, those cells that could not follow LED frequencies above 10 Hz also exhibited lower firing rates in response to steady-state current injection. Moreover, steady-state somatic current injection into these dLGN-INs resulted in a linear relationship between the maximum light-evoked AP rates and the maximum AP rates observed with steady-state current injection (Fig. 3c). Next, we examined the impact of optogenetic GABA release from dLGN-INs on the excitability of the surrounding thalamic relay neurons (Fig. 3d). One millisecond light pulses delivered every second reliably evoked fast IPSCs onto dLGN relay neurons, which were blocked by gabazine (Fig. 3e). Recruitment of extrasynaptic δ-GABA$_A$Rs following GABA spillover from RTN axons has been reported to prolong phasic inhibition in relay neurons of the ventrobasal complex[41]. A similar phenomenon occurs in the dLGN, and application of 10 μM DS2, a positive allosteric modulator of δ subunit-containing GABA$_A$ receptors (δ-GABA$_A$Rs)[43–45], resulted in a slowing of the ChR2-evoked IPSC that are driven by dLGN-INs (Fig. 3f). In the example shown, the evoked IPSC decay was best described by a double-exponential function. Following DS2 application the $\tau_{fast}$ component was moderately increased from 12 to 17 ms but the $\tau_{slow}$ component increased more markedly from 69 to 637 ms. Across all cells there was a significant increase in the weighted decay constant ($\tau_w$) from 18 ± 3 ($n = 4$) to 23 ± 2 ms ($P < 0.05$; paired *t*-test). The scatter plot (Fig. 3f) illustrates how DS2 was capable of slowing the decay of the evoked IPSCs with no change in the 10–90% rise time.

At 1 Hz stimulation rates, the latency between the start of the light pulse and the evoked IPSC was 3.3 ± 0.4 ms, with a response probability of 0.9 ± 0.04 ($n = 5$). However, at higher stimulation rates the IPSCs became less reliable (Fig. 3g). For example, following 10 s of stimulation at 20 Hz the response probability was reduced to 0.6 ± 0.1 ($n = 5$). In contrast to the frequency-dependent reduction in IPSC response probability, the tonic conductance was clearly enhanced at higher stimulation rates. For example, at 5 Hz the holding current increased by 24 ± 7 pA ($n = 14$ cells) compared with 172 ± 54 pA at 20 Hz. The increase in holding current observed at high stimulation rates was completely abolished by application of gabazine; consistent with the presence of a tonic GABA$_A$ receptor-mediated conductance

in dLGN relay neurons[42]. To identify the type of GABA$_A$R responsible for generating this tonic conductance we applied the drug DS2. DS2 enhanced the tonic current in all cells tested ($P < 0.05$, *t*-test, $n = 4$), and the action of DS2 on the tonic current was greater when stimulation rates were increased. In contrast, DS2 did not alter the frequency-dependent reduction in IPSC response probability (Fig. 3h). Therefore, the ability of DS2 to enhance the tonic conductance does not appear to involve any presynaptic action on vesicular GABA release, but is consistent with the action of an allosteric modulator on extrasynaptic δ-GABA$_A$Rs expressed on thalamic relay neurons.

By exploiting the *Sox14* promoter we were able to demonstrate how GABA released from the dLGN-INs results in a frequency-dependent activation of extrasynaptic δ-GABA$_A$Rs. When monitoring changes in membrane voltage in the presence of optogenetic GABA release from dLGN-INs we observed a switch from transient IPSPs at low stimulation frequencies to a steady-state hyperpolarization at higher stimulation frequencies (Fig. 3i). In all five cells examined the membrane hyperpolarized close to the estimated chloride reversal potential of − 68 mV consistent with the activation of a tonic GABA$_A$ receptor-mediated conductance of the type shown in Fig. 3g.

**No evidence for a thalamic origin of dLGN-INs.** The discovery that *Sox14* expression within the dLGN defines anatomically and functionally virtually all local interneurons implies that the current model for the ontogeny of this inhibitory cell type needs reassessing. In fact, dLGN-INs were previously hypothesized to originate in the telencephalic ganglionic eminence or ventral thalamus (p3)[26] and, more recently, described as a *Otx2*-dependent *Sox14⁻* lineage originating within p3 (ref. 25). We have extensively characterized neuronal precursors in the forebrain of $Sox14^{Gfp/+}$ knock-in mice and could not detect any *Sox14⁺* lineage in either of the two aforementioned territories. The presence of *Sox14⁺* neurons in the dLGN may be explained either by incoming migration from outside the nucleus or *de novo Sox14* expression. To test these hypotheses, we observed the progressive appearance of *Gfp⁺* cells in the dLGN of $Sox14^{Gfp/+}$ mice between P0 and P5 (Fig. 4a) and examined their morphology. A migratory morphology could be established for 42%, 40% and 31% of the total *Gfp⁺* dLGN population at P0 ($n = 64$ cells), P1 ($n = 211$ cells) and P2 ($n = 170$ cells), respectively; at subsequent days some *Gfp⁺* cells displayed morphological features of differentiated neurons, hence analysis was restricted to the P0–P2 window. The absence of weakly expressing *Gfp⁺* cells taken together with the abundance of migratory morphology in the *Gfp⁺* population suggests that active migration, rather than *de novo* activation of the *Sox14* promoter, is the reason behind the appearance of *Gfp⁺* cells in the dLGN. To estimate whether directionality can be established at population level, we mapped the leading process orientation of *Gfp⁺* neurons against the dLGN dorso-ventral and latero-medial axes (Fig. 4b). This was expressed both as raw angle and in its dorso-ventral and latero-medial components given by the sine and cosine, respectively (Fig. 4c). This analysis reveals that there is no obvious ventro-dorsally oriented migratory population, even at the onset of migration (P0), suggesting that a major contribution from the Th-R is unlikely. Instead, a predominant dorso-lateral to ventro-medial orientation of leading processes suggests a dorsal origin for dLGN-INs (Fig. 4c) (P0: $P < 0.025$; P1 and P2: $P < 10^{-3}$; one-sample Wilcoxon signed rank test).

Although the identification of a leading process and migratory morphology on fixed tissue is a good indication of on-going tangential migration, a definitive proof is the direct observation of active migration in the live tissue. Hence, we performed postnatal

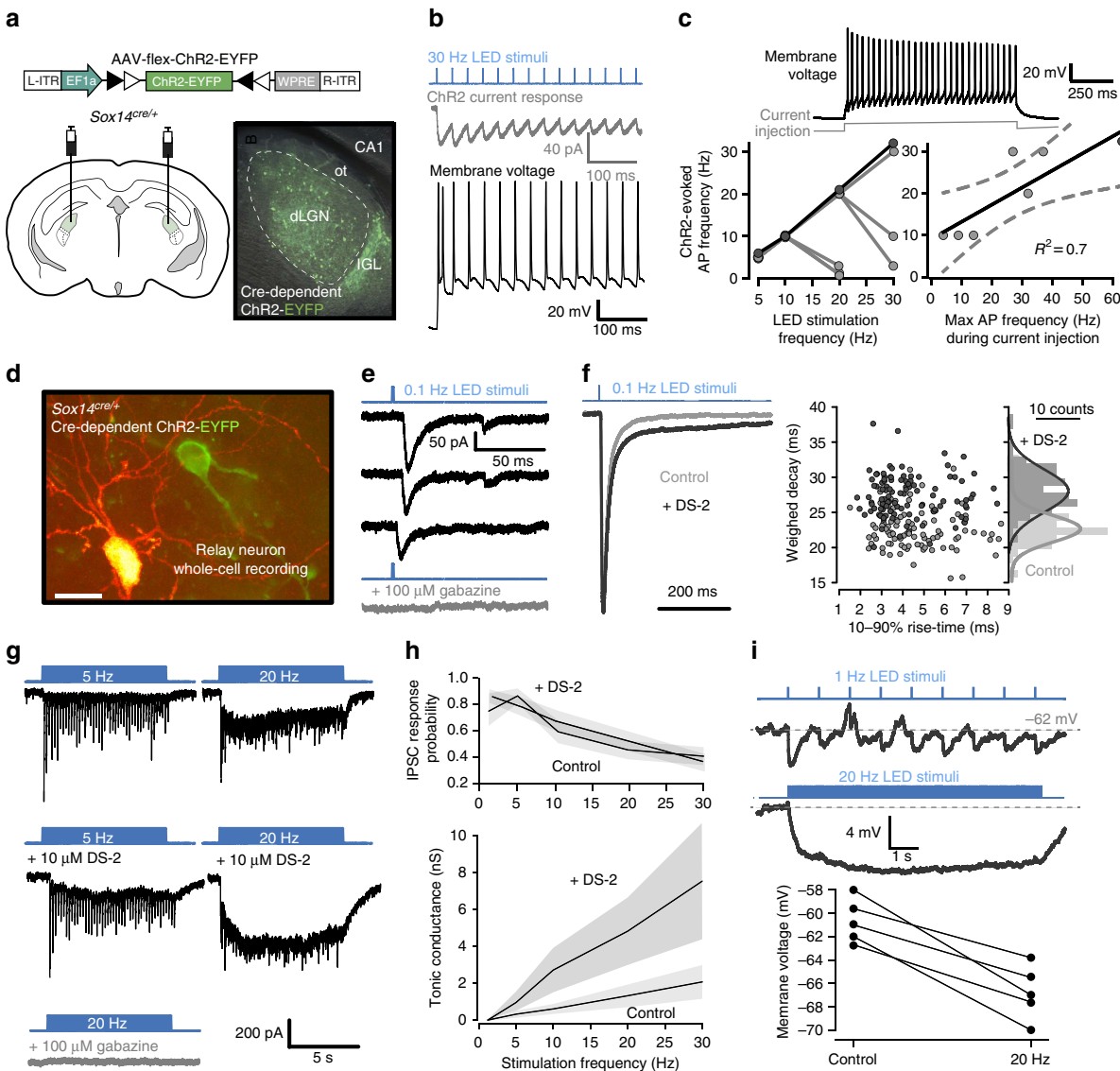

**Figure 3 | Sox14$^+$ interneurons contribute to phasic and tonic inhibition in the dLGN.** (**a**) Flex switch strategy for delivering ChR2-EYFP into Sox14$^{cre/+}$ neurons illustrating the injection strategy. Image of ChR2-EYFP distribution in the LGN of an acute slice preparation. (**b**) Voltage (black) and current traces (grey) shown in response to brief 1 ms blue light pulses (blue) in a ChR2-expressing Sox14$^+$ neuron. (**c**) The voltage trace obtained during current injection is from the cell shown in **b**. The plot on the left demonstrates the ChR2-evoked AP frequency at all LED stimulation rates with black filled symbols highlighting data from the cell shown in **b**. The doublet elicited with the initial photocurrent raised the expected AP frequency for this cell. The scatter plot compares the maximum AP frequency measured during current injection with the maximum AP frequency elicited by photocurrents. (**d**) Image showing a ChR2-eYFP-expressing cell (green) adjacent to a filled thalamic relay neuron (red). Scale bar, 30 μm. (**e**) Current traces recorded from a thalamic relay neuron during optogenetic stimulation. The black traces show light-evoked IPSCs. The bottom grey trace is recorded with all GABA$_A$ receptors blocked. (**f**) Average ChR2-evoked IPSC recorded before (grey trace) and after DS2 application (black trace). The scatter plot illustrates the relationship between IPSC rise and decay for evoked IPSCs recorded from a thalamic relay neuron. (**g**) The top current traces were recorded from a thalamic relay neuron during optogenetic stimulation at 5 and 20 Hz. The current traces recorded from the same thalamic relay neuron are shown in the presence of 10 μM DS2. The bottom grey trace shows the gabazine block at 20 Hz stimulation rates. (**h**) Quantification of tonic conductance and IPSC response probability in the presence and absence of DS2. The shaded areas show s.e.m. (**i**) Voltage trace from a thalamic relay neuron during brief light pulses (blue) delivered at 1 and 20 Hz. Scatter plot quantifies the change in membrane voltage during 20 Hz optogenetic stimulation in all cells examined.

time-lapse imaging on acute coronal sections at P0.5 and monitored the movement of Gfp$^+$ neurons over a 26 h period ex vivo. At P0.5 the dLGN is still largely devoid of Gfp$^+$ neurons, but a stream of ventrally oriented Gfp$^+$ neurons can be found in the region immediately dorsal to the dLGN. In the course of the following 26 h, several neurons from this region were seen entering the dLGN (Fig. 4d and Supplementary Movies 2 and 3). The migratory behaviour of Gfp$^+$ neurons in time-lapse movies correlates with higher intensity of endogenous fluorescence.

Notably, the live imaging demonstrates the presence of few Sox14$^{high}$ neurons in the IGL/vLGN and 1 of these enters the dLGN during the imaging period. This observation is consistent with the hypothesis that a small fraction of this descending population overshoots the dLGN and reaches the IGL/vLGN (Figs 6d and 7d) or may subsequently refine their position and re-enter the dLGN.

In summary, the migratory morphology and behaviour of Sox14$^+$ dLGN cells in fixed and ex vivo samples, strongly suggest

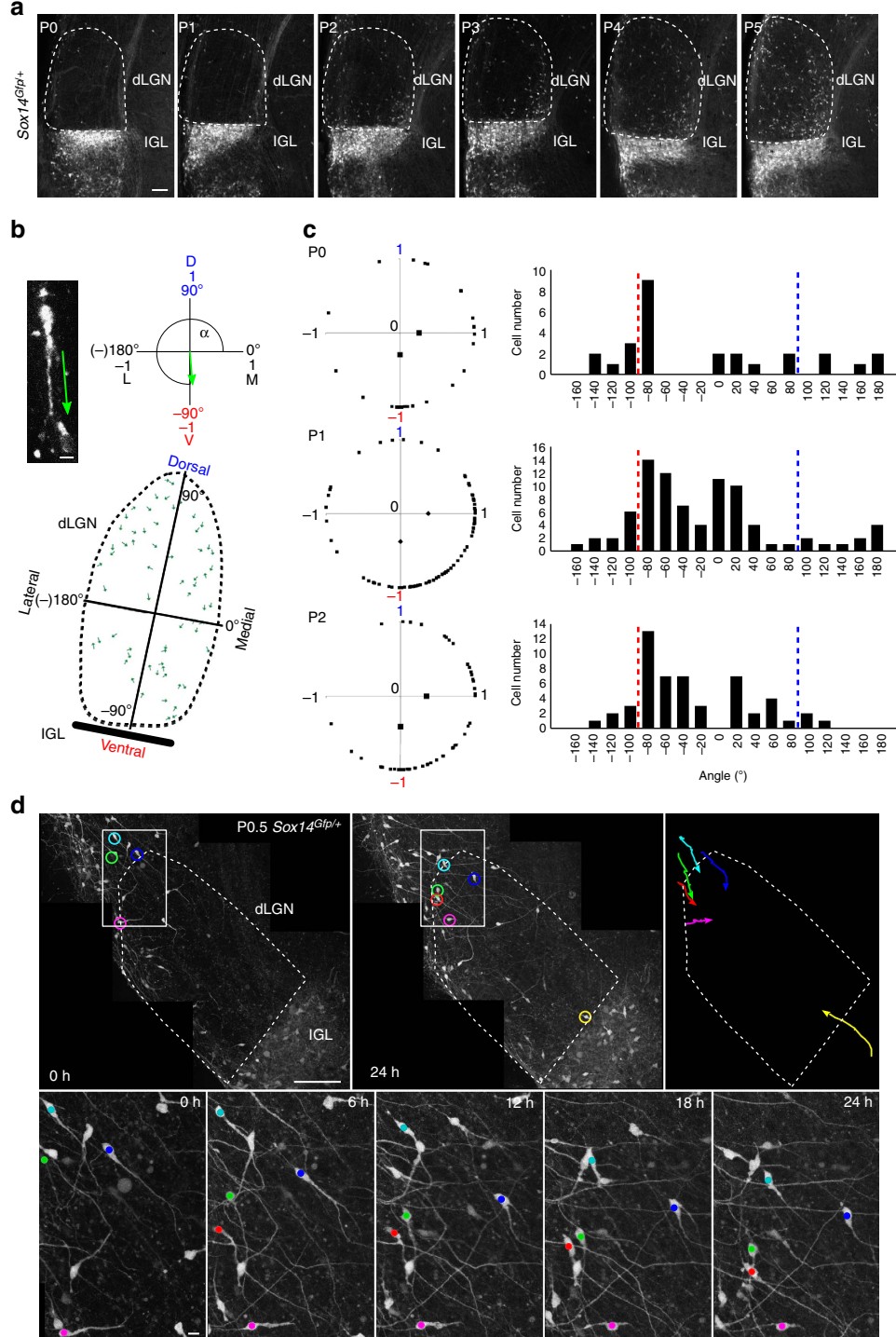

**Figure 4 | Most Sox14+ neurons migrate into the dLGN in a dorsal to ventral direction.** Cellular morphology and migratory behaviour of *Sox14+* neurons in the fixed and *ex vivo Sox14Gfp/+* dLGN. (**a**) Confocal imaging of *Sox14+* neurons in the dLGN from P0 to P5, showing the progressive spread of GABAergic interneurons in the relay dLGN. Scale bar, 100 μm. (**b**) Methodology used for population-wide analysis of the migratory morphology of Gfp+ neurons in the *Sox14Gfp/+* dLGN: each green unit depicts the leading process orientation. Its angular components relative to the dorso-ventral and latero-medial axes of the dLGN can be determined. Scale bar, 10 μm. (**c**) Quantification of the orientation of leading processes at population level at P0, P1 and P2: respective histograms displaying their angular distribution in 20° bins, with a maximum along the downward dorso-ventral axis (red dotted line) and lack of any obvious upwardly oriented (blue dotted line) population. Black dots on the plots of angular components are population means, which are significantly different from 0 at each developmental time point (P0: $P < 0.025$; P2–3 $P < 10^{-3}$; one-sample Wilcoxon signed rank test) and indicate a dominant dorso-ventral and latero-medial orientation ($n = 165$ cells from 1 brain per developmental stage). (**d**) Frame shots of *ex vivo* time-lapse imaging of an acute coronal section containing the *Sox14Gfp/+* dLGN at P0.5 and imaged over the following 26 h. Neurons that cross the boundaries of the dLGN are colour coded post acquisition for clarity (Supplementary Movies 2 and 3). Scale bar, 100 μm; and 10 μm for close-up images.

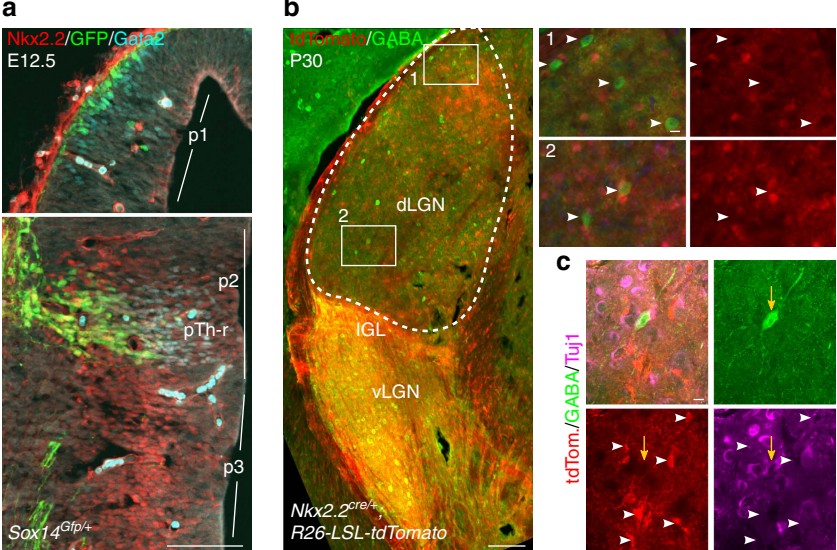

**Figure 5 | Fate mapping of *Nkx2.2* progenitors in the dLGN excludes GABA interneurons.** *Nkx2.2*$^{cre/+}$;*R26-LSL-tdTomato* conditional reporter expression was used to trace the fate of *Nkx2.2* progenitors in the thalamus. (**a**) IHC in the E12.5 brain of a *Sox14*$^{Gfp/+}$ embryo shows Nkx2.2 in progenitors of p3 and in the pTh-R domain of p2, which together define the presumptive GABA-rich vLGN and IGL territory, respectively (see also **b**). Nkx2.2 progenitors are not present in p1. Sequential activation of *Gata2* and *Sox14* defines a subset of Nkx2.2 progenitors in the pTh-R domain. Scale bar, 100 μm. (**b**) Adult brains (P30) of *Nkx2.2*$^{cre/+}$;*R26-LSL-tdTomato* mice reveal the known contribution of *Nkx2.2* progenitors to the vLGN and IGL. There is virtually no co-expression of GABA and tdTomato in the dLGN (white arrowheads in insets 1 and 2; out of 586 GABA$^+$ dLGN cells counted, 576 are negative for tdTomato and only 10 may have weak or unclassifiable tdTomato expression; N = 3 brains). Scale bars are 100 μm in the low-magnification image and 1 μm in insets. (**c**) The fate of *Nkx2.2* progenitors in the dLGN is largely non-neuronal, as indicated by weak to no staining for the pan-neuronal marker Tuj1 in tdTomato$^+$ cells of *Nkx2.2*$^{cre/+}$;*R26-LSL-tdTomato* mice (white arrowheads). A representative GABAergic, tdTomato-negative cell is also shown (yellow arrow). Scale bar, 1 μm.

that interneuron precursors reside in a dorsal region before entering the dLGN by active migration.

To conclusively rule out any contribution to dLGN-INs from p3 and p2 we took advantage of the expression of the *Nkx2.2* gene, which labels GABAergic fates in the presumptive vLGN and IGL[15,22] (Fig. 5a,b), to fate map *Nkx2.2* lineages with the previously described *Nkx2.2*$^{cre/+}$ knock-in line[46]. Fate mapping in adult brains from *Nkx2.2*$^{cre}$;*R26-LSL-tdTomato* mice reveals that, within the dLGN, the GABA$^+$ and the tdTomato$^+$ populations are mutually exclusive (Fig. 5b,c), with 98% of GABA$^+$ cells in the dLGN negative for tdTomato (*n* = 586 cells from 3 brains). Most tdTomato$^+$ cells are TuJ1$^-$ (Fig. 5c) and likely glia, in agreement with the known function of *Nkx2.2* in oligodendrocyte differentiation[47], as well as the role in neurogenesis. The evidence obtained so far is not supportive of a p3 or p2 origin for dLGN-INs, pointing instead at *Sox14*$^+$ and *Nkx2.2*$^-$ dorsal–caudal structures such as the pretectum (p1) and dorsal midbrain (Figs 5a, 7f).

**dLGN-INs are a Sox14$^+$ Otx2$^+$ lineage.** Conditional mutagenesis has shown that *Otx2* is required at the onset of tangential migration (P0) for the specification of dLGN-INs and that the dLGN of *Otx2*-ablated mice is devoid of virtually all inhibitory neurons[25]. We have therefore reasoned that the subpopulation of *Sox14*$^+$ precursors that gives rise to dLGN-INs must also be *Otx2*$^+$. Before the onset of tangential migration (E17.5), *Otx2*$^+$ neurons are present in the prethalamic side of the p3/p2 border, in the pretectum (p1) and in the adjacent superior colliculus (SC), but co-expression with *Sox14* is observed only within the superior colliculus and in isolated cells lining the pretectum and dLGN (arrows) (Fig. 6a). By the time dLGN-INs enter the dLGN (P3), *Sox14*$^+$ *Otx2*$^+$ double-positive neurons can be seen in the dLGN (Fig. 6b,d), in the superior colliculus and also forming a narrow

stream along the intervening outer region of the pretectum (arrows in Fig. 6c). The presence of a cluster of *Sox14*$^+$ *Otx2*$^+$ double-positive precursors in the superior colliculus just before birth that extends towards the dLGN is suggestive of a tectal origin for dLGN-INs. Before the onset of tangential migration (E17.5) the pretectum contains many *Otx2*$^+$ neurons, but none is also *Sox14*$^+$, suggesting that p1 is an unlikely source of *Sox14*$^+$ *Otx2*$^+$ double-positive dLGN-INs. Although there was no obvious *Sox14*$^+$ *Otx2*$^+$ double-positive population in the immature vLGN (E17.5-P3), some isolated double-positive cells were observed there at P3 (Fig. 6d,b), these may be migrating dLGN-INs that have overshot the dLGN and entered the vLGN (Fig. 7d). In summary, expression and loss-of-function data for *Otx2* and *Sox14* help to refine the ontogeny of dLGN-INs to a *Sox14*$^+$ *Otx2*$^+$ double-positive precursor subtype within the largely non-overlapping domains of *Otx2*$^+$ and *Sox14*$^+$ cells, mapping the likely location of dLGN-IN precursors to the superior colliculus (Fig. 6e).

**Tectal Sox14 precursors migrate into the dLGN.** To directly test the possibility that *Sox14*$^+$ interneurons migrate into the dLGN via a dorso-caudal to ventral route from the superior colliculus, we performed *in utero* labelling of *Sox14*$^+$ tectal precursor cells. We have injected through the uterine wall directly in the superior colliculus of E15.5 *Sox14*$^{cre/+}$ embryos a *cre*-dependent AAV expressing a red fluorescent protein (EF1a-flex-tdTomato). The dorso-caudal end of the telencephalic folds was taken as a landmark and the needle inserted just caudal to that. We then examined the location of labelled cells in juvenile mice at P14 (Fig. 7e). As expected, the injection site in the superior colliculus exhibited strong fluorescent signal (Fig. 7a), but no fluorescent labelling could be detected in more rostral pretectal territory, where a high number of *Sox14*$^+$ neurons reside (Fig. 7b,c).

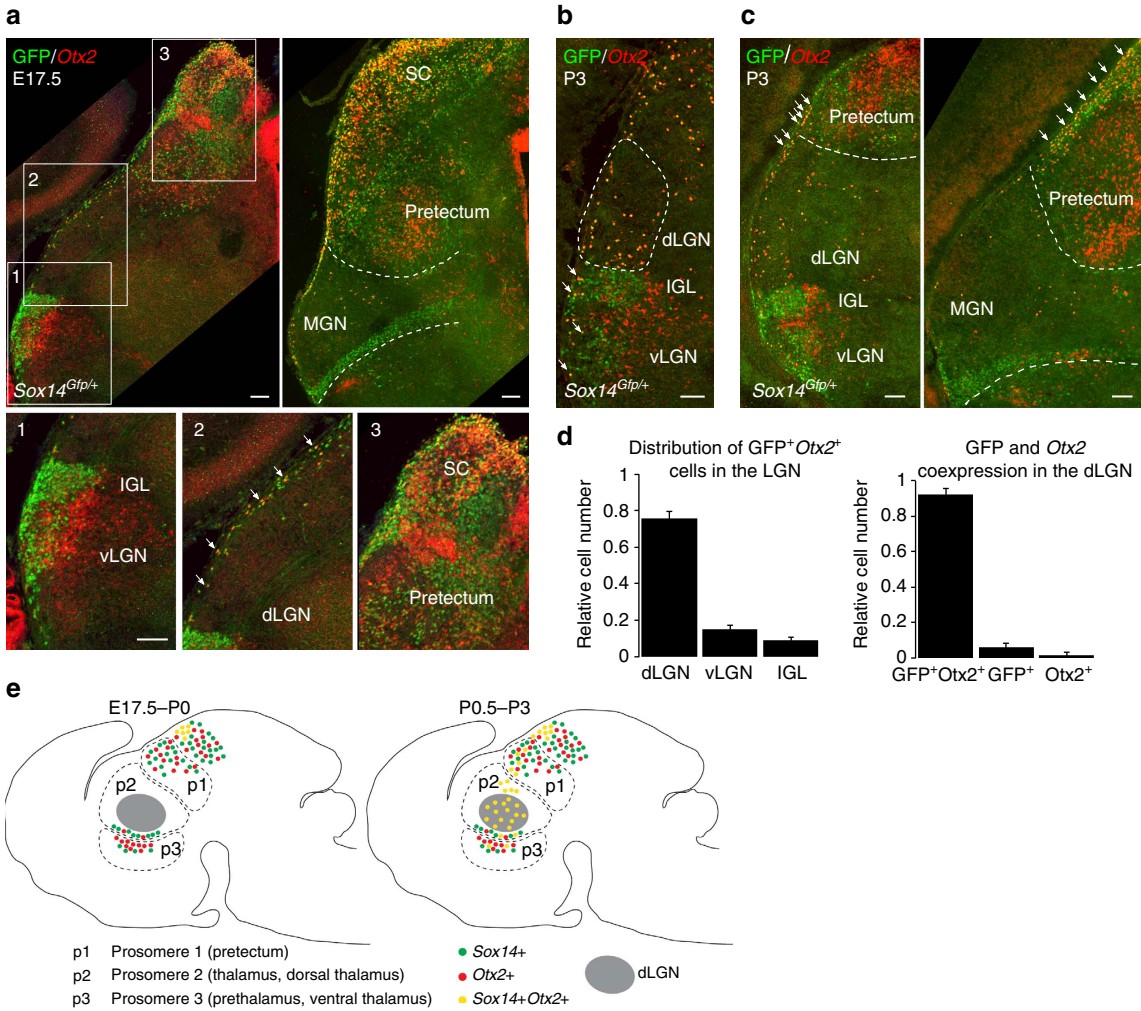

**Figure 6 | *Sox14* and *Otx2* are co-expressed in the embryonic tectum and both define the dLGN-IN population.** ISH for *Otx2* and IHC for GFP showing the location of double-positive *Sox14*[+]*Otx2*[+] cells before (E17.5) and after (P3) birth in *Sox14*[Gfp/ +] brains. (**a**) At E17.5 Gfp[+] and *Otx2*[+] neurons occupy adjacent domains of the vLGN (inset 1) and of the pretectum (inset 3). *Sox14*[+]*Otx2*[+] double-positive neurons are present in the outermost region of the superior colliculus (SC) and forming a stream of cells that extends from the SC towards the dLGN along the surface of the brain (white arrows in inset 2). (**b**) At P3 *Sox14*[+]*Otx2*[+] double-positive neurons are clearly visible in the dLGN and scattered in other thalamic nuclei. Furthermore, some double-positive neurons are also present in the IGL and vLGN (white arrows) and quantified in **d**. (**c**) The stream of *Sox14*[+]*Otx2*[+] double-positive neurons lining the brain at the level of pretectum and thalamus is highlighted with white arrows. (**d**) Quantification of the relative distribution of *Sox14*[+]*Otx2*[+] double-positive neurons in the three domains of the LGN (dLGN: 75.96 ± 3.71; vLGN: 14.93 ± 2.18; IGL: 9.10 ± 1.75; %, mean ± s.e.m.; $n = 531$ cells from 3 brains) and the relative numbers of double-positive and single-positive neurons in the dLGN (Gfp[+]Otx2[+]: 92.15 ± 0.47; Gfp[+]: 6.23 ± 0.36; Otx2[+]: 1.62 ± 0.81; %, mean ± s.e.; $n = 651$ cells from 3 brains). (**e**) Schematic summary of the distribution of *Sox14*[+]*Otx2*[+] double-positive and single-positive neurons across the thalamic (p3 and p2), pretectal (p1) and SC territories, before and after birth. Scale bars, 100 μm.

Instead, and in agreement with our hypothesis, the dLGN contained scattered labelled cells (Fig. 7b). Higher-resolution imaging revealed their polar morphology, consistent with them being inhibitory interneurons (Fig. 7b).

Importantly, at the time of injection and until birth, the dLGN is devoid of *Sox14*[+] cells, hence the fluorescently labelled neurons present in the nucleus at P14 must have entered it via tangential migration from the injected area. The possibility that the injected AAV solution may have spread within the extracellular space to reach the thalamus can be ruled out as no labelled cells were observed in *Sox14*[+] regions located between the injection site and the thalamus, such as the anterior pretectum (Fig. 7b,c). Therefore, we conclude that *Sox14* precursor cells migrating from the dorsal midbrain are the source of dLGN-INs.

Reminiscent of the few *Otx2*[+]*Sox14*[+] neurons observed in the IGL/vLGN (Fig. 6d), we have found some tdTomato-labelled

tectal neurons reaching the vLGN ( < 20% of total tdTomato-labelled LGN cells; $n = 145$). The high degree of correlation ($R^2 = 0.98$; Fig. 7d) in the distribution of tectal-derived *Sox14* neurons across the LGN subdomains that have been either labelled by *Sox14Otx2* co-expression or by *cre*-dependent AAV infection, further supports the conclusion that double-positive *Otx2*[+]*Sox14*[+] precursors reside in the dorsal midbrain before birth and migrate ventrally to seed the dLGN with local interneurons. Whilst most neurons terminate their migration in the dLGN, a smaller fraction continues migrating to reach the vLGN.

*Sox14* is not expressed in neuronal progenitors, hence the remote possibility exists whereby dLGN-IN precursors reside transiently in the dorsal midbrain between E15.5 and the time of birth, but are actually born elsewhere. To test whether dorsal midbrain progenitors contribute to dLGN-INs, we first

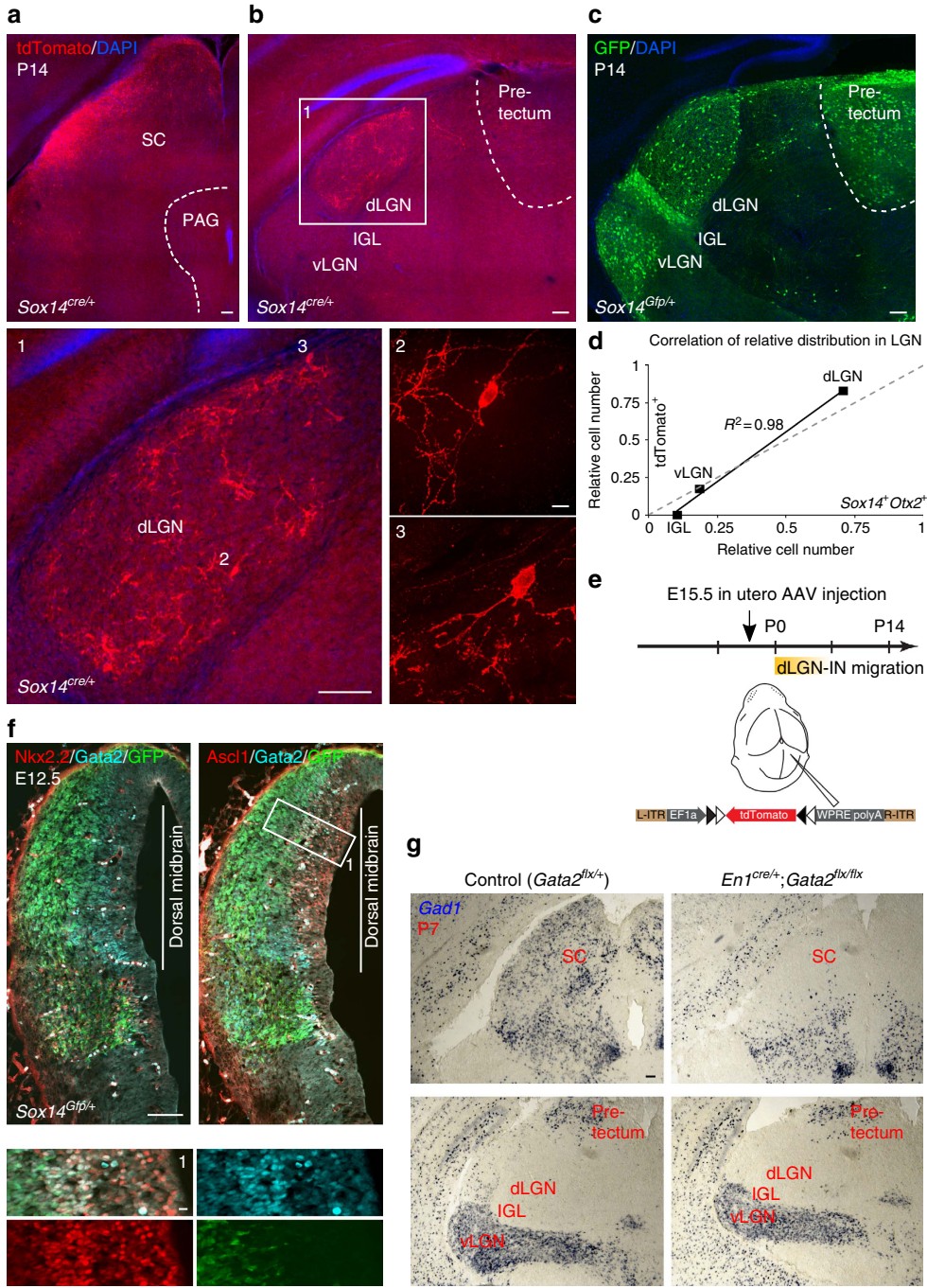

**Figure 7 | Fate mapping of tectal *Sox14*[+] precursors.** Labelling of *Sox14*[+] precursors via *in utero* focal injection of *AAV2/1-EF1α-DIO-tdTomato* virus into the SC of *Sox14*[cre/+] embryos at E15.5. The time line of the experiment is summarized in **e**. (**a**) At P14, the *cre*-dependent expression of tdTomato labels the site of injection in the outer-most region of the left SC. Scale bar, 100 μm. (**b**) Coronal sections at thalamic level reveal an enrichment of tdTomato-expressing cells within the boundaries of the dLGN. Scale bars, 100 μm. Higher-magnification confocal imaging (insets 2 and 3) reveals the typical morphology of local Ins; scale bar, 10 μm. (**c**) The location of *Sox14*[+] neurons in a comparative section from an age-matched *Sox14*[Gfp/+] brain illustrates the expected labelling pattern if the injected AAV-containing solution had passively spread from the site of injection to the thalamus. Scale bar, 100 μm. (**d**) Although not visible in **b**, in some brains tdTomato-labelled cells could be detected in the vLGN. The distribution of tdTomato-labelled *Sox14*[+] tectal precursors in the LGN subdomains is highly correlated to that of *Sox14*[+] *Otx2*[+] neurons in the same regions ($R^2 = 0.98$). (**e**) Time line of the *in utero* AAV injection and indicative view of the morphological landmarks used to select the unilateral injection site on the SC. (**f**) Neurogenesis in the E12.5 *Sox14*[Gfp/+] dorsal midbrain displaying the sequential activation of Ascl1 (Mash1), Gata2 and Sox14 (also see inset 1), and lack of Nkx2.2, which is instead expressed in a Sox14[+] ventral midbrain domain. Scale bars, 100 μm, and 10 μm for the inset. (**g**) Detection of the *Gad1* transcript in P7 control and *Gata2* conditional knockout brains. The *En1*[cre/+] driver selectively recombines the floxed (flx) *Gata2* allele in the midbrain, but not in more rostral forebrain structures. The marked reduction of *Gad1*[+] neurons in the SC is accompanied by a similarly strong reduction of *Gad1*[+] cells in the dLGN. In contrast, the density of *Gad1*[+] neurons in the intervening pretectum, as well vLGN/IGL remains grossly normal. Scale bar, 100 μm.

established the time of dLGN-IN genesis via 5-bromo-2′-deoxyuridine (BrdU) incorporation into S-phase genomic DNA of proliferating neural stem cells over the E10.5–E13.5 window of embryonic development, in $Sox14^{Gfp/+}$ mice. This birth-dating experiment revealed that more than 90% of dLGN-INs are born between E10.5–E13.5 with a peak of neurogenesis at days E11.5 and E12.5 combined (38.7 ± 0.3 and 37.7 ± 5.2, respectively; %, mean ± standard error; $n = 3$ brains per time point) (Supplementary Fig. 1). We then introduced a plasmid driving constitutive enhanced green fluorescent protein (eGFP) expression in the proliferative zone of the dorsal midbrain at E12.5, via in utero electroporation and analysed the electroporated brains at P3 (Supplementary Fig. 1). Electroporated brains showed extensive eGFP labelling across the tectum with some eGFP expression retained in the glia cells that form the ependymal layer lining the midbrain aqueduct, but not those lining the diencephalic third ventricle (Supplementary Fig. 1). The restricted labelling of midbrain ependymal cells at P3 was taken as a consequence of the specific targeting of dorsal midbrain progenitors at E12.5. Importantly, eGFP$^+$ cells were also detected in the thalamus (Supplementary Fig. 1), predominantly in the dLGN (50%), but also in the vLGN/IGL complex (11%) and in the remainder of the thalamus (39%)—principally in the lateral posterior and ventrobasal thalamic nuclei. Immunodetection of GABA confirmed the inhibitory identity of the dLGN GFP$^+$ cells (Supplementary Fig. 1).

All evidence so far weights strongly in favour of a tectal origin for all dLGN-INs. Yet, our in utero dorsal midbrain labelling of progenitors and precursors cannot categorically exclude the possibility of unintentional labelling in the nearby diencephalic pretectal (p1) territory.

To exclude the pretectum as a dLGN-IN-contributing territory, we took advantage of the known requirement for $Gata2$ in inhibitory fate identity and used the $En1^{cre/+};Gata2^{flx/flx}$ conditional mutant to specifically ablate the GABAergic fate in the dorsal midbrain while preserving $Gata2$ expression and inhibitory neurogenesis in all diencephalic compartments, including the pretectum (p1, p2 and p3)[48]. $Gata2$ is required for $Sox14$ expression within $Ascl1^+$ inhibitory lineages across the midbrain and diencephalic p1, p2 and p3, and is transiently co-expressed with $Sox14$ on cell cycle exit (Fig. 7f)[19] (K. Achim, A. Kirjavainen and J.P., unpublished). Remarkably, the dLGN of $En1^{cre/+};Gata2^{flx/flx}$ shows a 91% reduction of GABAergic cells ($En1^{cre/+};Gata2^{flx/flx}$ dLGN: 101 $Gad1^+$ cells; control $Gata2^{flx/+}$ dLGN: 1,090 $Gad1^+$ cells. On the basis of 25 and 26 sections, respectively, covering the entire rostro-caudal span of the nucleus. $N = 2$ brains per genotype.), which correlates with the similarly strong reduction of inhibitory neurons in the dorsal midbrain (Fig. 7g). In contrast, other GABA-rich diencephalic territories, as well as the ventral midbrain, are grossly unaffected (Fig. 7g). Hence a conditional mutagenesis strategy that disrupts the acquisition of the inhibitory fate in the midbrain but not in the diencephalon, confirms that the tectum and not the nearby pretectum, is the main source of dLGN-INs.

## Discussion

We provide the first experimental evidence for a tectal origin of thalamic interneurons and redefined dLGN-INs as an $Nkx2.2^-$ $Gata2^+ Otx2^+ Sox14^+$ lineage, requiring $Gata2$, $Otx2$ and $Sox14$ expression. Using optogenetics, we exploit this previously unappreciated genetic signature to demonstrate a frequency-dependent switch from a phasic to a tonic form of inhibition within the visual thalamus.

Ventro-dorsal oriented tangential migration is known to underlie the positioning of interneurons into the pre-existing excitatory pyramidal neuron organization of the neocortex[3], and a similar developmental scenario is considered to take place within the thalamus with interneurons migrating from the ventral (p3) region of the thalamus[26]. In support of this established textbook view[27], tangentially migrating GABAergic precursors that move in a rostro-caudal direction in the developing thalamus have been described[49,50], although the fate of these neurons does not appear to be local thalamic interneurons, but rather to coincide with an early wave of migrating pTh-R precursors, which peaks at E12.5–E13.5 and is also labelled by $Sox14$ expression[17]. This early wave of tangentially migrating $Sox14^+$ neurons contributes inhibitory cells to the perilateral habenula and the nucleus posterior limitans, at the dorsal and caudal edges of the thalamus, respectively[17]. The bulk of the GABAergic neurogenesis that takes place in the pTh-R will contribute to the IGL and part of the vLGN[15,17,19,22]. It has been observed that thalamic interneurons appear within the thalamus after birth[25,26]. We now show that none of the diencephalic progenitor domains (p1, p2 and p3) is a likely source of dLGN-INs. Our observation that dLGN-INs are an immigrant tectal population reinforces the important role of tangential migration[51] in seeding distant networks with GABAergic interneurons. In contrast to cortical interneurons, which arise and migrate within the boundaries of the telencephalon, we suggest that thalamic interneurons are not diencephalic, but born in an evolutionary and developmentally distinct brain region: the midbrain. Furthermore, the genetic programme leading to tangential migration must also be different, as $Arx$, a key regulator of tangential migration in cortical interneurons[52], is not expressed by $Sox14^+$ neurons.

Tangential migration is likely controlled by genetically encoded programmes of the type described in this paper, but maturation of the visual circuitry in the dLGN also requires presynaptic activity in the retinogeniculate pathway[53–57]. Recently, spontaneous retinal waves have been shown to influence the final positioning of dLGN-INs along the medio-lateral axis of the nucleus[25]. During this critical time window, postsynaptic activity is also likely to play an important role[58,59] and GABAergic interneurons may contribute to the functional maturation of visual circuits[60,61]. Although we did not report any gross abnormality in the segregation of retinal afferents in the dLGN of $Sox14$ knockout mice, more detailed future analysis is required to establish a role for dLGN-INs in visual circuit refinement.

Unpublished data from our laboratory suggest that other interneuron populations in the ventrobasal and lateroposterior thalamic nuclei are also strongly reduced in the $Gata2^{flx/flx};En1^{cre}$ and $Sox14$ knockout brains and are likely of mesencephalic origin. The relative abundance of inhibitory interneurons across various thalamic nuclei of different mammalian species may correlate with the relative complexity of thalamic circuits[62]. Future comparative work is required to establish whether the tectal origin of thalamic inhibitory interneurons is shared beyond the visual thalamus in carnivorans and primates.

Nevertheless, by exploiting the genetic signature of dLGN-INs we were able to demonstrate, using an optogenetic strategy, that $Sox14^+$ dLGN-INs generate both phasic and tonic forms of inhibition, but activation of extrasynaptic δ-GABA$_A$Rs was enhanced when dLGN-IN firing rates were high. Indeed, simultaneous paired recording between dLGN-INs and thalamic relay neurons was characterized by a dominance of extrasynaptic δ-GABA$_A$R activation. The low prevalence of AP-evoked IPSCS in these simultaneous recordings may, in part, be explained by the synaptic fatigue that is clearly observed with optogenetic stimulation of GABA release from dLGN-IN terminals. A previous study[41] has demonstrated that evoked IPSCs in the ventrobasal thalamus also contained a slow component that was mediated by extrasynaptic δ-GABA$_A$Rs. Other studies have

shown that genetic removal of synaptic GABA$_A$Rs from the ventrobasal thalamus did not disturb thalamocortical slow oscillations and sleep spindles[63]. The fact that dLGN-INs are capable of activating extrasynaptic δ-GABA$_A$Rs may explain why thalamocortical oscillations can be maintained in the absence of synaptic GABA$_A$ receptors.

Tonic inhibition of the type we describe in this study will be driven by visual input to the dLGN, whereas in other regions of sensory thalamus the GABAergic drive will be generated solely from the surrounding RTN. The identity of the retinal ganglion cell (RGC) types that make synaptic connections with $Sox14^+$ dLGN-INs will need to be established, but an intriguing scenario would see them innervated by luminance-detecting intrinsically photosensitive RGCs[64] providing a potential mechanism to regulate excitability of relay neurons to environmental luminance. The tools we have developed for genetically targeting $Sox14^+$ dLGN-INs will help us identify the RGC types that excite thalamic interneurons.

## Methods

**Animals.** The $Sox14$::$Cre$-targeting vector was constructed by replacing the $Sox14$-coding sequence with a cassette consisting of a 5' splice substrate followed by the $cre$ gene and a bovine growth hormone polyadenylation signal ($bGHpA$). A neomycin gene cassette was cloned downstream of the $bGHpA$ signal for positive selection in embryonic stem cells. The $Sox14$::$Cre$ targeting vector was electroporated into mouse embryonic stem cells (129sv/ev), selected with G418 and homologous recombinants were identified by Southern blot analysis. The targeted embryonic stem cells were injected into blastocysts, and chimeras were crossed to C57BL/6J mice. The PGKneo cassette was removed after crossing with the $FLPeR$ mice[65]. The $Sox14^{Gfp/+}$ (refs 17,66) and $Sox14^{cre/+}$ mouse lines were maintained in the C57Bl/6 background in the animal facilities of King's College London and housed under standard conditions. All experimental procedures described have received internal approval by the King's College London Ethical Committee and are covered by a UK Home Office Licence. The genetic background and age of mouse lines used are described in the relevant 'Results' and 'Methods' sections. Animals of both sexes were used to produce the data presented.

**Stereotaxic brain injections.** For optogenetics, $Sox14^{cre/+}$ P16 pups were anaesthetized with isoflurane and full anaesthesia maintained throughout the procedure. The head was fixed in a small animal stereotaxic frame (World Precision Instruments), the skull exposed via a 1 cm longitudinal skin incision and the initial dLGN coordinates defined as $y$: 1.7; $x$: 2.4; $z$: 3.0 from lambda, then optimized for each litter. $Sox14^{cre/+}$ mice were injected bilaterally using borosilicate glass needles connected to an air injector system (Narishige). With an approximate volume of 0.2–0.3 µl of $rAAV2$-$EF1a$-$DIO$-$hChR2(H134R)$-$EYFP$-$WPRE$ (Addgene plasmid 20298). Electrophysiological experiments were performed 2 weeks after surgery.

**BrdU labelling and detection.** Wild-type C57BL/6 dams were mated with heterozygous $Sox14^{Gfp/+}$ males. BrdU (Sigma, B5002) was administered intraperitoneally to pregnant mice (0.2 mg g$^{-1}$) at 09:00 between E10.5 and 13.5, estimated from the occurrence of a vaginal plug (morning of the day the plug was detected was designated E0.5). Gfp heterozygous pups were perfused at P7 with 4% paraformaldehyde (PFA) and treated for cryosectioning as above. For BrdU/Sox14 double labelling, 40 µm floating sections were cut, followed by denaturation with 1 M HCl in H$_2$O at 45 °C for 30 min and neutralization with three washes of phosphate-buffered saline (PBS; pH 7.4) for 10 min. Sections were then blocked in 2% normal goat serum (NGS), 0.3% Triton-X, in PBS for 1 h at room temperature, and probed with rat antiBrdU (1:200, in 2% NGS, 0.3% Triton-X, in PBS, OBT0030CX Bio-Rad) and chicken anti-GFP (Ab13970 Abcam) primary antibodies at 4 °C for 48 h, followed by fluorescent Goat Alexa-568 anti-rat and Alexa-488 anti-chicken (A11039, Invitrogen) secondary antibodies. Counting of BrdU$^+$ and Gfp$^+$ double-positive cells in the $Sox14^{Gfp/+}$ dLGN used every second section, covering the whole span of the dLGN ($n = 3$ brains per developmental stage).

**In utero brain injections and electroporation.** To maximize litter size and the chances of survival of the litter, we crossed $Sox14^{cre/+}$ males to CD1 dams. The CD1 pregnant dam was maintained under full anaesthesia with isoflurane throughout the procedure. Incision was made through the skin and the abdominal wall to expose the uterine horns. An approximate volume of 0.2–5 µl of $rAAV2$-$EF1a$-$DIO$-$tdTomato$-$WPRE$ produced as serotype 1 was injected using borosilicate glass needles connected to an air injector system (Narishige). A stereomicroscope was used to target the injection to the embryonic tectum of E15.5 embryos. Mice were killed at P14 and genotyped for presence of the $cre$ allele. The brains were

sectioned at 60–80 µm and images were acquired with a spinning disk confocal microscope (Eclipse Ni-E Upright, Nikon). Three $Sox14^{cre/+}$ embryos were injected in the tectum, and on postnatal analysis displayed selective enrichment of tdTomato-labelled cells in the LGN. For control three $Sox14^{cre/+}$ embryos were injected in the hypothalamus. On postnatal analysis these showed labelling of hypothalamic $Sox14^+$ neurons, but not the thalamus. Counting of tdTomato-expressing cells in the LGN was done on every second section, covering the whole span of the LGN (145 cells from 2 brains). Focal labelling of cre-expressing cells was consistently observed ipsilateral to the injection site, which excludes the possibility that labelling occurs via accidental leakage of the virus in the brain ventricles. For in utero electroporations, reporter and $Cre$-dependent plasmids (pCAG-eGFP at 0.25 µg µl$^{-1}$ and pEF1aDIOdsRed2WPRE at 0.75 µg µm$^{-1}$) were injected in the midbrain ventricle at E12.5. Five voltage pulses (37 mV, 50 ms) were then applied using Electroporator CUY21 (Nepa Gene) and tweezers with asymmetric round plate platinum electrodes (1–5 mm diameter; CUY650P1–5 Nepa Gene) to allow for more focal targeting. The brains were collected at P3 and processed and analysed as above ($n = 2$ brains).

**Anterograde tracing of the optic tract.** P19 $Sox14^{Gfp/+}$ mice were anaesthetized with isoflurane and a small incision was performed through the cornea of one eye to facilitate access to the intravitreum. The eye cavity was filled with Alexa-594-conjugated CTb at 1 µg µl$^{-1}$ in PBS (Life Technologies) using a pulled borosilicate glass needle and an injector pump (Picospritzer III). Three days later animals were perfused with 4% PFA/PBS under terminal anaesthesia. Brains were postfixed in 4% PFA solution at 4 °C overnight and further processed for cryosectioning and immunohistochemistry (IHC) to amplify the Gfp signal.

**IHC and RNA in situ hybridization.** Mice were transcardially perfused with 4% PFA in PBS and the brains postfixed at 4 °C overnight for IHC (longer for in situ hybridization (ISH)). Brains were equilibrated in 30% sucrose/PBS, embedded in OCT freezing compound and cut on a cryotome at 80–100 µm for IHC, or 20 µm for ISH and IHC on embryonic tissue. IHC was performed on floating sections using 7% goat serum/PBS with 0.3% TritonX-100 as blocking and antibody-binding solution. The following primary antibodies were incubated on sections overnight at 4 °C: rabbit aOtx1–2 (1:1,200, ab21990 Abcam); chick aGfp (1:10,000, ab13970 Abcam); rabbit aCasp3 (1:300, ab13847 Abcam); rabbit adsRed (1:200 Clontech); mouse aTUJ1 (1:300, MMS-435 Covance); rabbit aGata2 (1:200, H-116 Santa Cruz); mouse aMash1 (Ascl1) (1:100, 556604 BD Biosciences); mouse aNkx2.2 (1:50, 74.5A5 DSHB); and rabbit aGABA (1:2,000, A2052 Sigma). All secondary antibodies were Alexa-conjugated goat aIgG (1:500, Life Technologies). ISH was performed as previously described[17]. The $Otx2$ antisense RNA probe was transcribed in vitro from a full-length cDNA template (IMAGE: 4527414). The $Gad1$ probe was a kind gift from John Rubenstein, UCSF, San Francisco, USA. On satisfactory colour reaction with Fast Red substrate, sections were immunoreacted with a chick antiGfp antibody for 48 h at 4 °C, and fluorescent images acquired using a spinning disk confocal microscope (Eclipse Ni-E Upright, Nikon). Counting of $Sox14^+$ and $Otx2^+$ cells in the $Sox14^{Gfp/+}$ LGN at P3 was done on every fifth section (20 µm thickness), covering the whole span of the LGN ($n = 3$ brains). Counting of $Gad1^+$ cells in the $En1^{cre/+}$;$Gata2^{flx/flx}$ or control ($Gata2^{flx/+}$) dLGN at P7 was done in the same way ($n = 2$ brains per genotype). Counting of GABA$^+$tdTomato$^+$ cells in the $Nkx2.2^{cre/+}$;$R26$-$LSL$-$tdTomato$ dLGN at P30 used every second section (60 µm thickness), covering the whole extent of the dLGN ($n = 3$ brains).

**Ex vivo time-lapse imaging.** Brains of P0.5 $Sox14^{Gfp/+}$ mice were dissected on ice and embedded in 4% low–melting-temperature agarose. The 250 µm-thick sections were cut with vibratome (Leica VT 1200S) in ice-cold sucrose solution (all in mM): 70 sucrose; 86 NaCl; 4 KCl; 1 NaH$_2$PO$_4$; 1.4 MgCl$_2$; 26 NaHCO$_3$; 1 CaCl$_2$; and 24 glucose, and held in ice-cold Krebs buffer post sectioning (all in mM): 12.6 NaCl; 0.25 KCl; 0.12 NaH$_2$PO$_4$; 0.12 MgCl$_2$; 0.21 CaCl$_2$; 25 NaHCO$_3$; 11 glucose, and supplemented with HEPES (Invitrogen), penicillin, streptomycin and gentamicin (Invitrogen). Sections containing dLGN were placed on Millicell culture filters (Millipore, 0.4 µm, 30 mm diameter), floating on pre-warmed MEM medium (Gibco) supplemented with glucose, FCS, penicillin and streptomycin (Invitrogen). After 1 h the medium was replaced with Neurobasal (Gibco), supplemented with B27 (Gibco), glutamine (Invitrogen), glucose, penicillin and streptomycin (Invitrogen). After 16 h in culture the sections were imaged with an inverted laser scanning spectral confocal microscope (Leica TCS SP2 RS) using water-immersion × 25 objective, and maintained at 37 °C with 95%O$_2$/5%CO$_2$. Images were taken every 60 min over a 26 h period and analysed for cell movement in FIJI using the Pairwise stitching and Manual Tracking plugins.

**Quantification of migrating neurons.** For each $Sox14^{Gfp/+}$ brain (P0–P2), z-stack images of the dLGN with interplanar separation of 1.8 µm, spanning on average 50 µm, were acquired with a spinning disk confocal microscope (Eclipse Ni-E Upright, Nikon), taken at × 20 magnification. Every second dLGN-containing section was imaged for each brain (3–4 sections per brain). The acquired z-stacks were analysed for the migratory morphology of dLGN $Sox14^+$ cells by marking the orientation of their leading processes using the Icy software arrow

tool. Leading processes were identifiable in 165 out of 445 (37%) $Gfp^+$ dLGN cells. To quantify their orientation, the angles of the arrows were determined in FIJI relative to the $x$ axis of the dLGN (width). The latter was defined by taking the $x$ axis as parallel to the base of the nucleus and positive in the medial direction, and the $y$ axis as the midline perpendicular to the base and positive in the dorsal direction. The angles were plotted on a histogram, using 20° bins. In addition, the component of the orientation along the dorso-ventral and latero-medial directions for each leading process was determined by taking the sine and cosine of the angles, respectively. This produces indices between 1 and −1 for each direction. In the latero-medial axis, an index of 1 corresponds to purely medial orientation, and of −1 to purely lateral orientation. In the dorso-ventral direction, an index of 1 corresponds to purely dorsal orientation, and −1 to purely ventral orientation. The dorso-ventral and latero-medial indices for each leading process were plotted on a scatter plot in the dLGN $x$–$y$ coordinate system across all sections, for each developmental stage, using Microsoft Excel. The average value of each index is also shown for each plot, where an average of 0 would suggest equal representation of all orientations. As the indices were not normally distributed, the Wilcoxon signed rank test was used to establish that average indices were significantly different from 0.

**Three-dimensional-image reconstruction of the dLGN.** Z-stack images, with an interplanar separation of 11 μm, of the $sox14^{Gfp/+}$ P3 dLGN were acquired with a spinning disk confocal microscope (Eclipse Ni-E Upright, Nikon), using a × 10 objective. Every dLGN-containing section was imaged and the acquired z-stacks were aligned and assembled in rostral to caudal progression in FIJI using the TrakEM2 plugin.

**Acute slice preparation and whole-cell patch clamp recording.** Brain slices were obtained from both male and female mice at postnatal 21–30 days, which were killed by cervical dislocation followed by decapitation. The brain was rapidly removed from the skull and immersed in ice-cold slicing solution composed of (in mM) the following: 85 NaCl; 2.5 KCl; 1 CaCl₂; 4 MgCl₂; 1.25 NaH₂PO₄; 26 NaHCO₃; 75 sucrose; and 25 glucose, pH 7.4, when bubbled with 95% O₂ and 5% CO₂. Coronal brain slices (250 μm thickness) were cut (Campden Instruments) and immediately transferred to a holding chamber containing slicing artificial cerebrospinal fluid (ACSF) aerated with 95% O₂/5% CO₂. Slices were then transferred to a 37 °C heat block for 10 min, after which the slicing ACSF was exchanged for recording ACSF (in mM: NaCl 125; KCl 2.5; CaCl₂ 2; MgCl₂ 1; NaH₂PO₄ 1.25; NaHCO₃ 26; and glucose 11, pH 7.4, when bubbled with 95% O₂ and 5% CO₂). The slices were incubated in the recording ACSF at room temperature for at least another 30 min before electrophysiological recordings. The recording chamber was perfused with carbogen-saturated ACSF with a flow rate of 2–5 ml min⁻¹ using a gravity perfusion system. Patch pipettes were fabricated with a two-step vertical puller (Narishige, PC-10), with tip resistance at 4–8 MΩ when back-filled with internal solution. For whole-cell voltage-clamp recordings, the patch pipettes were filled with Cs-based internal solution containing following (in mM): CsCl 140; NaCl 4, CaCl₂ 0.5; HEPES 10; EGTA 5; and Mg-ATP 2; pH 7.3, adjusted with CsOH. For whole-cell current-clamp recordings, we used internal solution containing the following (in mM): 145 K-gluconate; 4 NaCl; 5 KCl; 0.5 CaCl₂; 5 EGTA; 10 HEPES; 4 Mg-ATP; and 5 sucrose, pH 7.3, adjusted with KOH. Fixed-stage upright microscope (Olympus BX51W1) fitted with a water-immersion objective (Olympus, × 60) and a non-immersion objective (Zeiss, × 1.25) was used to visualize the neurons in slices. Whole-cell recordings were performed with a Multiclamp 700B amplifier (Molecular Devices) at room temperature. The analogue output was low-pass filtered at 10 kHz, digitized at 20 kHz using a BNC-2120 device (National Instruments), and analysed using WINWCP and WINEDR software (John Dempster, University of Strathclyde, UK). In most cases, biocytin (2 mg ml⁻¹) was added to the internal solution and patch pipette was slowly pulled away from the neuron after whole-cell recording. The slice tissue was then preserved in 4% PFA for over 48 h. To start the immunostaining process, PFA was washed off with ice cold PBS three times, 10 min each time. Slices were then blocked and permeabilized with 5% donkey serum and 0.2% TritonX-100 in PBS-based solution at room temperature for 1–2 h. After washing with PBS for 10 min, slices were submerged in 2 mg ml⁻¹ streptavidin, Alexa Fluor 555 Conjugate (Life Technologies)-added PBS solution with 1% donkey serum and 0.2% Triton X-100 for 3–4 h at room temperature. Subsequently, slices were washed again in PBS for three times, 10 min each, and then mounted on slides with mounting medium (H-1000, Vectashield).

**Optogenetic stimulation of Sox14⁺ dLGN-INs.** A blue (470 nm) collimated LED (M470L3-C1, Thorlabs) was mounted to the back of the microscope, and focused through the objective lens. The optical power emitted by our × 63 water-immersion lens increased linearly to a maximum power of 70 μW mm⁻² at 1 V. A 1 ms, 1 V brief pulse protocol gave rise to a transient response that peaked at 40 μW mm⁻² with a 10–90% rise time of 0.73 ms and a decay constant of 9.65 ms; as determined from a single exponential fit. In a series of calibration experiments we observed that CHR2-evoked responses were not detected when the illumination spot was moved ∼200 μm from the recorded cell, corresponding to an illumination area of 0.03 mm².

**Quantification of tonic and phasic conductance changes.** All-point histograms were constructed from 1 s epochs of the holding current recorded under conditions, where the tonic conductance was not present before chloride loading, present in control conditions, blocked using GABA_A receptor antagonists or enhanced by the allosteric modulator DS2. A single Gaussian function was used to estimate the mean holding current and the tonic conductance was calculated from the driving force according to the relationship; $G_{tonic}$ = change in mean holding current driving force. The quality of a linear regression was established using the coefficient of determination or $R^2$, with values closer to 1 indicating a high degree of confidence in the regression model. All statistical hypothesis testing was performed assuming normal data distributions unless otherwise stated.

**Data availability.** The data that support the findings of this study are available on request from the corresponding authors (S.G.B. and A.D.).

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

## Acknowledgements

We are grateful to Oscar Marin and Lynette Lim at the MRC Centre for Developmental Neurobiology, King's College London for their technical help with live time-lapse imaging, to Diana Garofalo and Dina Balderes from the Sussel lab for providing *Nkx2.2^cre;R26-SLS-tdTomato* brains and to Anna Kirjavainen for perfusion fixation of *En1^cre;Gata2^flx* brains used in this study. This research was funded by BBSRC Grant BB/L020068/1 to A.D. and a Wellcome Trust Grant WT094211MA to S.G.B. and W.W.

## Author contributions

All authors contributed extensively to the work discussing the results and commenting on the manuscript at all stages. A.D. and S.G.B. instigated this work and wrote the paper; P.J., Z.Y. and A.D. performed the experiments and analysed data; X.Y., J.P. and L.Z. provided additional materials.

## Additional information

**Competing financial interests:** The authors declare no competing financial interests.

