## [Peer Review File · Nature Communications]

Reviewers' comments:

Reviewer #1 (Remarks to the Author):

This paper documents on novel origin of inhibitory interneurons in the visual thalamus of mice. The transcription factor Sox14 is expressed embryonically in tectal neural precursors, and these cells give rise to the majority of inhibitory neurons that reside within the post-natal IgN.

A combination of genetic tagging (with gfp), electrophysiology, and optogenetics provides a host of results that support these findings. The paper will be of interest to developmental biologists that study the visual system, and others in related fields.

Most of the results are quite convincing at face value, and the overall presentation could be greatly improved through clarifications and corrections.

A detailed list of issues follows:

1) There is some confusion regarding tonic vs phasic current. As Herd et al (cited) have shown, extrasynaptic receptors on thalamic relay neurons are recruited during evoked synaptic release. This results in recruitment of extrasynaptic receptors and an increase in the duration of the evoked, phasic response. This is exactly what is reported here with the evoked responses resulting from optogenetic activation of sox14 cells. There is not a single experiment here that directly assesses tonic inhibition, per se.

2) The existence of synaptic triads in the mouse IgN is somewhat controversial. Bikcford et al, 2010, say that neuroanatomical markers for triads are "often" associated with presumed interneuron terminals, and then only in adult animals. The association the authors propose between triadic terminals and mGlu activation in justifying their DHPG experiment is unnecessary when it is probably sufficient to simply refer to IN terminals.

3) Given that interneurons mainly contribute to feed forward inhibition of LGN relay neurons via retinal inputs, and this through local dendrodendritic interactions that don't necessarily involve global interneuron activation, the authors should justify the use of global optogenetic activation, which presumably results in non-local actions simultaneously at all dendrodendritic synapses. This is critical because the authors would like to make conclusions about recruitment of "extrasynaptic" receptors by sox14 neurons, and such recruitment obviously depends on pooling as would result from cooperation between nearby synapses.

4) Several minor points are not made very clearly or directly in the paper. For example,

A) the case for differing morphology of sox14+ vs sox14- IgN neurons is not very strong, only depicting a single example of a raw fluorescence image of the former but not the latter. A simple reconstruction (NeuroLucida) of example cells of each type would make the point much more clearly.

B) Several figures include "scatter plots" which by definition are bivariate. This does not appear to be the case for figure 2e.

2) There are several overstatements and or illogical mistakes that have crept in to the text and should

be removed. A careful read will identify these, and a partial list includes

a) Figure 2 legend: "...absence of GABA positive ...", when a reduction (admittedly at ~90%), not absence is noted.

b) Results page 5: there is a presumption (not stated) that absence of Sox14 cells results in no compensatory change in the output of RTN. In the absence of any understanding of potential compensations in the system, then the conclusive statement that "RTN terminals provide at least 30% of .. GABA" is unwarranted.

c) Either GABA is gone from LGN in sox14 deleted animals (gaba and gad are undetectable in lgn, page 4, results) or GABA function is still maintained, presumably from RTN fibers and terminals (Figure 2F,G). Both GAD and GABA are likely highly expressed in inhibitory terminals. This discrepancy needs to be addressed.

D) Maximal firing rates are reported, but these to my eyes are not maximal, as the firing rates in figure 1 continue to rise with increasing depolarizations, and a maximum is not reached within the range of stimuli. Furthermore the stimuli are not even provided, such that this plot actually plots two non-independent variables (Vm, spike rate) vs each other. In fact the independent variable in such an experiment is the depolarizing stimulus. Finally, in this analysis, reporting of membrane voltage here doesn't make sense, as clearly the Vm is changing continuously during the spike train.

E) I am very confused by the authors' use of the term "holding current" (page 6, results), which normally refers to the steady, resting state of the cell, and would seem inappropriate for description of dynamically changing ionic conductances.

Reviewer #2 (Remarks to the Author):

The manuscript by Jager et al. provides developmental evidence for the unexpected origin of GABAergic interneurons in the dLGN nucleus of the thalamus that appear in early postnatal stage in mice. A major tool is a Sox14-EGFP knock-in mice, with which they show all the GABA-positive cells in postnatal dLGN are Sox14-EGFP positive and that Sox14 is required for the appearance of EGFP-positive cells there. In addition, the authors analyzed the morphology of EGFP-positive neurons as well as time-lapse imaging showing a robust caudal-to-rostral migration of EGFP-positive neurons into the dLGN nucleus. Furthermore, using Sox14-Cre mice and a localized injection of Cre-dependent AAV expressing tdTomato into the midbrain, they demonstrated the contribution of midbrain cells to the dLGN nucleus. In addition to the above developmental data, the authors also used electrophysiology and optogenetics to characterize the properties of GABA neurons in the dLGN.

As for the developmental part of the work, it provides novel finding of the extra-diencephalic origin of inhibitory neurons in the thalamus and has a broad impact pertaining to the general concept of neuronal migration. Therefore, the significance of the work is high. One major criticism is the lack of strong genetic evidence that supports the mesencephalic origin of these neurons. Since gene expression could be very dynamic during development, it is difficult to exclude the possibility that p3, p2 (pTH-R) or p1 (pretectum) contributes to GABAergic cells in dLGN. For example, although pretectal neurons may not co-express Otx2 and Sox14 in embryonic stage, they may do so as they migrate caudally towards the dLGN. In addition, the possibility of de novo expression of Sox14 from the p3-derived cells is hard to exclude solely based on the migratory morphology. Additional data on Cre mice (crossed with reporters) specific to progenitor cells in each of these prosomeres (or their combination) would significantly strengthen the authors' claim of the origin of dLGN interneurons.

Additional comments:

The description of current consensus about the origin of the dLGN interneurons may not be accurate. Edward Jones' book ("The Thalamus") lists possibilities of the origins of dLGN interneurons, but he does not make a conclusive comments due to the lack of experimental evidence in mice. The only

report that clearly goes against the current finding is Goldberg et al., (2014) in which they claim that dLGN interneurons come from the prethalamic cell lineage that once reside in vLGN before birth. Therefore, no previous reports provided evidence that dLGN interneurons are derived from the thalamus. More accurate description of the present finding would be the first demonstration of extra-diencephalic origin of dLGN interneurons, which by itself is very novel from a broad perspective of cell migration in the brain. If the authors are to go by the prosomeric model, it is less confusing to define the thalamus as p2-derived structure (except the dLGN interneurons), not including the prethalamus, a p3-derivative, in this category.

Reviewer #3 (Remarks to the Author):

Review of Jager et al., 2016 Nature Communications

Jager and colleagues have set out to answer the important question of the developmental origin of interneurons in the thalamic dorsal lateral geniculate nucleus (dLGN). To answer this question they utilise immunohistochemistry, fluorescence imaging, optogenetics and electrophysiology in a Sox14GFP/+ mouse line. The major finding of the study is a non-thalamic developmental origin for interneurons in the mouse dLGN. The authors nicely demonstrate that the precursor cells that mature into dLGN interneurons migrate into this nucleus from the tectum during early postnatal development.

I feel that this manuscript can be broadly divided into two components; the developmental origin of dLGN interneurons and the electrophysiology of interneurons in the Sox14GFP/+ mouse line. The former represents the more novel physiological aspect of the manuscript whereas the latter validates a new optogenetic tool (Sox14 cre mouse) but largely confirms physiological findings previously reported elsewhere (see below).

The authors nicely demonstrate that interneurons in the dLGN migrate in a dorsal to ventral direction during early postnatal development using ex vivo time lapse imaging of GFP expressing neurons. I found this data convincing and indicative that dLGN interneuron precursors do not migrate from the IGL/vLGN. Furthermore, I thought the in utero injection experiments to determine the tectal origin of Sox14 precursors were cleverly conceived and sufficiently convincing to support their conclusions.

The identification of the necessity of Sox14 expression for the appearance of functional interneurons in the dLGN is very important and has enabled the authors to develop a new tool (a Sox14 cre mouse) that will undoubtedly be extremely useful in the future study of the role of dLGN interneurons both in vitro and in vivo. I commend the authors for this work.

Overall, I found this manuscript to be well written and the experiments well thought out and executed. The demonstration of an extrathalamic (tectal) developmental origin for interneurons in the dLGN to my knowledge is novel and highly interesting.

However, I feel that a number of issues arise that need to be addressed in order to make the manuscript suitable for publication in Nature Communications.

Major concerns

In the abstract the authors state 'Using optogenetics we show that at high firing rates (1) Sox14+Otx2+ dLGN-INs generate a powerful form of tonic inhibition (2) that will regulate the gain of thalamic relay neurons (3) through recruitment of extrasynaptic high-affinity GABA_A receptors'. There

are three significant issues (see corresponding numbers above and below) with this statement.

1) I have serious concerns regarding the interpretation of the optogenetic experiments that I think require major revision involving reanalysis of data and/or rewriting of the manuscript.

Principally, the problem rests with the fact that the firing of the interneuron does not follow the stimulation frequency at higher stimulus frequencies. At 10 Hz (as shown in Figure 3C) the firing of the interneuron follows the light stimulation frequency despite the relatively long (~100 ms) photocurrents evoked by each stimulus. However, at 30 Hz and I suspect all frequencies above 10 Hz (Figure 3C), the interneuron firing clearly does not follow the stimulation with a significant number of blue light pulses not producing an action potential. At higher frequencies the stimulation is acting more like a step-function opsin due to the summation of the photocurrents (i.e. like injecting a long square depolarizing current step) and producing 'tonic' like firing in the interneuron. For example, the ~500 ms section trace shown in Figure 3C has stimulation at 30 Hz but only 8 spikes (~16 Hz) and appears to show considerable spike frequency adaptation.

Consequently although the authors describe 'high firing rates' in interneurons we do not actually know what frequency the cells are firing at. The optimum solution would be to reanalyse the data/perform further experiments to determine the actual mean interneuron firing rates obtained by optogenetic stimulation at each frequency and to show this in the manuscript.

These problems somewhat undermine their claims for a frequency-dependent switch from phasic to tonic inhibition (pg 13 ln 7) since it is unclear how the increasing stimulation frequency actually relates to increasing interneuron firing and GABA release.

This also invalidates the interpretation of the data shown in Figure 3H on 'release probability' since at higher frequencies the reduction is most likely due to the discrepancy between the number of light stimulation pulses and the number of spikes the interneuron actually fires rather than a reduction in neurotransmitter release probability following an action potential.

2) The demonstration of frequency dependent modulation of extrasynaptic δ -containing GABA receptors in TC neurons by dLGN interneurons is not novel. This has already been demonstrated by Errington et al., 2011 who showed using pharmacological and electrical stimulation (of retinal inputs to the triad) that activation of mGluRs on interneuron dendrites increases IPSC frequency and tonic current in TC neurons. This study largely confirms these earlier findings using an optogenetic approach. I think it is necessary for the authors to acknowledge this in more detail. Currently the manuscript gives the impression that this is the first demonstration of modulation of tonic GABAergic inhibition in TC neurons by interneurons in the dLGN, which it is not.

3) I do not see any evidence presented to support the idea that tonic inhibition mediated by GABA release from interneurons 'will regulate the gain of thalamic relay neurons.' Could the authors please revise this appropriately to reflect the fact that this is speculation.

Minor points

1) To distinguish putative interneurons from thalamocortical neurons in the dLGN it would be useful for the authors to comment on the presence or absence of low threshold spike bursts in these cell types since all TC neurons in dLGN produce LTS bursts whereas interneurons do not. If the authors have this data available I think it would help to support the electrophysiological characterization of the cells.

Furthermore, interneurons in dLGN have slower membrane time constants - do the authors also see

longer membrane time constants in GFP+ cells in the Sox14GFP/+ mouse.

In my opinion these are more obvious electrophysiological identifiers (along with input resistance as the authors have shown) for interneurons versus TC cells in dLGN than the I-V relationship or spike firing frequency profile.

This is not a critical point and I am convinced based on the electrophysiological presented that the GFP+ cells are interneurons but I think these suggestions could improve the manuscript.

2) In Figure 1F both the GFP+ and GFP- cells seems to have quite depolarized resting membrane potentials (injected current 0 pA) for these types of cells in vitro. In the Methods I could not find out if the membrane potentials quoted are corrected for the liquid junction potential. Could the authors please include this in a revised manuscript.

3) The reduction in GABAergic inhibition in TC neurons in the absence of GAD67 expressing dLGN interneurons in Sox14GFP/GFP mice, although not particularly surprising, indicated, at least in brain slices under basal conditions, that interneurons may provide the lion's share of inhibition in dLGN. I think it would be sensible, however, where the authors give the figure of 30% (pg 5, ln 25) of inhibition coming from TRN that they reiterate that this is in the slice and that the in vivo situation could be markedly different.

4) Interpretation of the DHPG experiment is complicated. The authors suggest that dLGN interneuron terminals are 'confined within the glomerular arrangement of the thalamic triad' (pg 6 ln 2). It is my understanding that this may be true for dendrodendritic F2 terminals but is not true for conventional axonal F1 terminals. The action of DHPG on dLGN interneurons is mediated by F2 terminals since F1 terminals do not express mGluRs. Consequently the lack of effect of DHPG in Sox14GFP/GFP could be interpreted to stem from a selective disruption of dendritic signalling rather than a complete loss of functional interneurons, especially since some interneurons do remain in the dLGN of Sox14GFP/GFP mice (Figure 2E and pg 4 ln 19). I personally think this is unlikely given the other data presented but I feel some rewriting and acknowledgment of the previous literature here would help.

5) Pg 3 ln 13 - change 'that they contain GABA neurotransmitter' to 'they contain the neurotransmitter GABA.'

6) It is not immediately obvious why the authors perform anterograde labelling using injection of Alexa-594 conjugated CTb. I understand it is to identify the visual thalamic nuclei but I think a sentence explaining why they have done it to make it immediately clear to the reader would help.

7) Pg 4 ln 23 - the authors say that 'Sox14+ neurons in the dLGN are the only resident inhibitory cell type'. I think this conclusion needs to be toned down slightly given that in Sox14 knockout mice there are still some GFP positive cells in the dLGN and expression of Gad1 and the presence of GABA are 'virtually' but not completely undetectable. Also on pg 7 ln 17-18 the authors say 'virtually all local interneurons'. This is inconsistent with the statement above and the authors should try to be consistent throughout.

8) Statistics. I could not find any description of the decisions behind why particular statistical tests were used or which analysis software was used to perform the tests in the Results or Methods. I may have missed it but the authors should check and insert a section on statistics into the Methods if appropriate. Also check the figure legends as some did not describe what particular error bars on graphs represented as required by the journal. Finally, ensure consistency in descriptions (e.g. in Figure 2 - mean {plus minus} SEM is used whereas in Figure 5 we see average {plus minus} SE).

9) The epifluorescence image in Figure 1D is not particularly clear. Could the authors image the cells

using a confocal or 2-photon microscope to present a clearer image? Furthermore, if the authors are contrasting the somatodendritic morphology of the GFP+ (putative interneurons) and GFP- (putative TC cells) to aid in cell classification they need to show a comparison for both cell types and ideally some quantitative measurement (i.e. soma size/dendritic length).

Finally, I found the description given of the relative morphology of interneurons and TC cells confusing. In both rat and mouse dLGN I would not consider the morphology of interneurons to be compact. They typically have much longer dendrites than TC neurons. The reason for the higher input resistance and lower membrane capacitance is due to the smaller soma size and thinner dendrites (which I think is what the authors mean) but I would not consider the somatodendritic morphology of these interneurons to be compact either physically or electrotonically. I would request that the authors carefully reword this section to clarify this point.

Reviewer #4 (Remarks to the Author)

The Jager et al., study on "Tectal derived Sox14+Otx2+ interneurons contribute to phasic and tonic inhibition in the visual thalamus" is an important and timely contribution to the field. It describes a previously unappreciated population of Sox14+Otx2+ dLGN-INS control thalamic relay neuron excitability. The paper describes the mechanistic details how these neurons regulate the gain of thalamic relay neurons through recruitment of extrasynaptic high-affinity GABAA receptors.

The paper provides the first experimental evidence for a tectal origin of thalamic interneurons and redefined dLGN-INS by their expression of Otx2 and Sox14.

The authors describe that Most Sox14+ neurons migrate into the dLGN in a dorsal to ventral direction. It would be interesting to know more about the:

- A: Clonal relations and exact lineage. Some future clonal analysis should be done (un due course).
- B: Examine the possibility that retinal activity regulates this migration to dLGN in enucleation or pharmacological manipulation studies.

The paper raises important questions about the utility of the prosomeric model in the finer connectivity in the diencephalon. Perhaps this is not the place to go into the details of these issues, but I hope the authors shall consider contributing a more detailed review to this specific issue.

I have little to criticize. The paper was put together with care and attention. All figures tell a story and the conclusions are sound. The paper opens up several important developmental and evolutionary/comparative questions and I have no doubt that it will attract general readership.

Note: key text modifications in the revised manuscript in response to reviewers 1 and 3 are in blue, those in response to reviewer 2 are in red.

Reviewer #1 (Remarks to the Author):

- 1) There is some confusion regarding tonic vs phasic current. As Herd et al (cited) have shown, extrasynaptic receptors on thalamic relay neurons are recruited during evoked synaptic release. This results in recruitment of extrasynaptic receptors and an increase in the duration of the evoked, phasic response. This is exactly what is reported here with the evoked responses resulting from optogenetic activation of sox14 cells. There is not a single experiment here that directly assesses tonic inhibition, per se.

We now express data dealing with changes in holding current as changes in tonic conductance (see modified methods section and revised plots in Figure 1 and 3). Moreover, in the revised figure 1 we illustrate how the tonic conductance measurement was obtained from single Gaussian fits to all-point histograms constructed from 1 second current epochs ($G_{\text{tonic}} = \text{change in mean holding current} \div \text{driving force}$).

We agree with the reviewers' interpretation that Herd et al (2013) have demonstrated that evoked IPSCs were prolonged by DS-2 application following electrical stimulation within the thalamus. On page 8, line 14 we now discuss the fact that our data focuses on the possibility that GABA release from local Sox14 interneurons is similarly able to recruit extrasynaptic GABA-A receptors. This issue is addressed further in the discussion section on page 19.

- 2) The existence of synaptic triads in the mouse lgn is somewhat controversial. Bikcford et al, 2010, say that neuroanatomical markers for triads are "often" associated with presumed interneuron terminals, and then only in adult animals. The association the authors propose between triadic terminals and mGlu activation in justifying their DHPG experiment is unnecessary when it is probably sufficient to simply refer to IN terminals.

Although we do feel there is evidence to suggest that synaptic triads are a feature of the mouse dLGN (see reviewer 3) we now simply refer to dLGN-IN axon terminals when discussing the actions of DHPG on page 7, line 4.

- 3) Given that interneurons mainly contribute to feed forward inhibition of LGN relay neurons via retinal inputs, and this through local dendrodendritic interactions that don't necessarily involve global interneuron activation, the authors should justify the use of global optogenetic activation, which presumably results in non-local actions simultaneously at all dendrodendritic synapses. This is critical because the authors would like to make conclusions about recruitment of "extrasynaptic" receptors by sox14 neurons, and such recruitment obviously depends on pooling as would result from cooperation between nearby synapses.

We have included a discussion of this important point in the methods section on page 23, line 24-30 of the revised manuscript. Briefly, blue LED light enters the slice through a 0.9 NA 63x water immersion objective lens giving an effective illumination area of 0.126 mm². This will cause activation of all dLGN-IN terminals in this area of the slice. We did a series of control experiments where we recorded from a thalamic

relay neuron and moved the illumination spot away from the recorded cell. Evoked GABA responses were completely lost once the illumination area was ~200 μm from the recorded cell; consistent with a limited area of illumination. We did not intend to imply that this mode of stimulation mimics physiological mechanisms of GABA release from dLGN-INs. We have attempted to clarify this point by including the statement "we examined the impact of optogenetic GABA release from dLGN-INs" on page 7, line 21-22. Moreover, inclusion of data from simultaneous recordings between a single interneuron and relay neurons also sheds light on this issue (see modified figure 1) and response to reviewer 3.

4) Several minor points are not made very clearly or directly in the paper. For example, A) the case for differing morphology of sox14+ vs sox14- lgn neurons is not very strong, only depicting a single example of a raw fluorescence image of the former but not the latter. A simple reconstruction (NeuroLucida) of example cells of each type would make the point much more clearly.

We agree with the reviewer and have included a full reconstruction of the Sox14 +ve dLGN-IN 1 and a Sox14 -ve relay neuron for comparison.

B) Several figures include "scatter plots" which by definition are bivariate. This does not appear to be the case for figure 2e.

Most scatter plots are indeed of bivariate data sets whereas the data in Figure 2e is not. We have modified the figure to avoid any confusion.

2) There are several overstatements and or illogical mistakes that have crept in to the text and should be removed. A careful read will identify these, and a partial list includes

a) Figure 2 legend: "...absence of GABA positive ...", when a reduction (admittedly at ~90%), not absence is noted.

Done

b) Results page 5: there is a presumption (not stated) that absence of Sox14 cells results in no compensatory change in the output of RTN. In the absence of any understanding of potential compensations in the system, then the conclusive statement that "RTN terminals provide at least 30% of .. GABA" is unwarranted.

This statement has been modified accordingly

c) Either GABA is gone from LGN in sox14 deleted animals (gaba and gad are undetectable in lgn, page 4, results) or GABA function is still maintained, presumably from RTN fibers and terminals (Figure 2F,G). Both GAD and GABA are likely highly expressed in inhibitory terminals. This discrepancy needs to be addressed.

Although GABA should be present in the remaining RTN terminals this signal is below the detection threshold of our imaging. This presumably reflects the small volume of axon terminals (sub-micron in diameter). The loss of GABA in the much larger soma volumes was observed. We would need EM analysis to confirm the presence of GABA in the remaining RTN terminals within the Sox14 knockout LGN. Our quantification of GAD was at the mRNA level and so does not report terminal expression of the enzyme. However, we totally agree that GABA and GAD will still be

present in RTN terminals in the Sox14 knockout and we have modified the text on page 6 to reflect this.

D) Maximal firing rates are reported, but these to my eyes are not maximal, as the firing rates in figure 1 continue to rise with increasing depolarizations, and a maximum is not reached within the range of stimuli. Furthermore the stimuli are not even provided, such that this plot actually plots two non-independent variables (Vm, spike rate) vs each other. In fact the independent variable in such an experiment is the depolarizing stimulus. Finally, in this analysis, reporting of membrane voltage here doesn't make sense, as clearly the Vm is changing continuously during the spike train.

What determines the firing rate of an interneuron is the Vm and current injection is often used to modify the Vm but the resting Vm is different for each cell. We therefore felt it was more appropriate to plot the relationship between Vm and firing rate. To avoid confusion, we have decided to remove this analysis from Figure 1 and instead we have compared the maximum AP frequency between steady-state current injection and optogenetic stimulation in dLGN-INs (see Figure 3C).

E) I am very confused by the authors' use of the term "holding current" (page 6, results), which normally refers to the steady, resting state of the cell, and would seem inappropriate for description of dynamically changing ionic conductances.

Our apologies for any confusion. Holding current is an accurate description of what is being measured using the voltage-clamp technique. However, to help clarify the relationship between holding current and tonic conductance we have modified figure 1 and elaborated on the methods section.

Reviewer #2 (Remarks to the Author):

The manuscript by Jager et al. provides developmental evidence for the unexpected origin of GABAergic interneurons in the dLGN nucleus of the thalamus that appear in early postnatal stage in mice. A major tool is a Sox14-EGFP knock-in mice, with which they show all the GABA-positive cells in postnatal dLGN are Sox14-EGFP positive and that Sox14 is required for the appearance of EGFP-positive cells there. In addition, the authors analyzed the morphology of EGFP-positive neurons as well as time-lapse imaging showing a robust caudal-to-rostral migration of EGFP-positive neurons into the dLGN nucleus. Furthermore, using Sox14-Cre mice and a localized injection of Cre-dependent AAV expressing tdTomato into the midbrain, they demonstrated the contribution of midbrain cells to the dLGN nucleus. In addition to the above developmental data, the authors also used electrophysiology and optogenetics to characterize the properties of GABA neurons in the dLGN.

As for the developmental part of the work, it provides novel finding of the extra-diencephalic origin of inhibitory neurons in the thalamus and has a broad impact pertaining to the general concept of neuronal migration. Therefore, the significance of the work is high. One major criticism is the lack of strong genetic evidence that supports the mesencephalic origin of these neurons. Since gene expression could be very dynamic during development, it is difficult to exclude the possibility that p3, p2 (pTH-R) or p1 (pretectum) contributes to GABAergic cells in dLGN. For example, although pretectal neurons may not co-express Otx2 and Sox14 in embryonic stage, they may do so as they migrate caudally towards the dLGN. In addition, the possibility of de novo expression of Sox14 from the p3-derived cells is hard to exclude solely based on the migratory morphology. Additional data on Cre mice (crossed with reporters) specific to progenitor cells in each of these prosomeres (or their combination) would significantly strengthen the authors' claim of the origin of dLGN interneurons. **We accept this criticism and have taken on board this reviewer's advice that new data with cre-mice, specific to the relevant progenitor domains, would strengthen our previous conclusions. We have now used:**

- a) *Nkx2.2^{cre/+};R26-LSL-tdTomato* to label the fate of GABA progenitors in p3 and p2 (pTh-R). Result: no contribution to dLGN-INs [New Figure 5].
- b) *En1^{cre/+};Gata2^{flx/flx}*, which ablates *Gata2* in the midbrain but not diencephalon and causes the loss of GABAergic identity in the dorsal midbrain only (leaving p3, p2 and p1 inhibitory neurogenesis intact). Result: dramatic reduction of dLGN-INs [updated Figure 7 –formerly Figure 6].

We are pleased that the analysis of these two cre-mediated approaches fully confirms and strengthens our previous model whereby thalamic interneurons are of tectal origin. Furthermore, we have now established the time of birth of dLGN-INs and have targeted by focal in utero electroporation the dorsal midbrain progenitor domain at the time when dLGN-INs are born [new Supplementary Figure 1], which confirms that progenitors born in this territory contribute GABAergic cells to the dLGN.

The manuscript now includes novel data on the requirement for *Gata2* within the midbrain to support dLGN-IN differentiation. We suggest to remove “*Sox14+Otx2*” from the title for 2 reasons: a) a more accurate description now should read “*Nkx2.2-Gata2+Sox14+Otx2*” which may affect readability of the title; b) the “*Sox14+Otx2*” may give the false impression that this is one of several subpopulations of dLGN-INs.

Additional comments:

The description of current consensus about the origin of the dLGN interneurons may not be accurate. Edward Jones' book ("The Thalamus") lists possibilities of the origins of dLGN interneurons, but he does not make a conclusive comments due to the lack of experimental evidence in mice. The only report that clearly goes against the current finding is Goldberg et al., (2014) in which they claim that dLGN interneurons come from the prethalamic cell lineage that once reside in vLGN before birth. Therefore, no previous reports provided evidence that dLGN interneurons are derived from the thalamus. More accurate description of the present finding would be the first demonstration of extra-diencephalic origin of dLGN interneurons, which by itself is very novel from a broad perspective of cell migration in the brain. If the authors are to go by the prosomeric model, it is less confusing to define the thalamus as p2-derived structure (except the dLGN interneurons), not including the prethalamus, a p3-derivative, in this category.

We have revised the wording in the text to reflect these points.

Abstract (page: 1, line: 22): replaced "extra-thalamic origin" with "extra-diencephalic".

Introduction (page: 2, line: 23): replaced "thalamic" with "prethalamic".

Discussion (page: 18, line: 5- 6): "p3" and "pTh-R" jointly replaced with "diencephalic".

Reviewer #3 (Remarks to the Author): Review of Jager et al., 2016 Nature Communications

Major concerns

1) I have serious concerns regarding the interpretation of the optogenetic experiments that I think ***require major revision involving reanalysis of data and/or rewriting of the manuscript.***

Principally, the problem rests with the fact that the firing of the interneuron does not follow the stimulation frequency at higher stimulus frequencies. At 10 Hz (as shown in Figure 3C) the firing of the interneuron follows the light stimulation frequency despite the relatively long (~100 ms) photocurrents evoked by each stimulus. However, at 30 Hz and I suspect all frequencies above 10 Hz (Figure 3C), the interneuron firing clearly does not follow the stimulation with a significant number of blue light pulses not producing an action potential. At higher frequencies the stimulation is acting more like a step-function opsin due to the summation of the photocurrents (i.e. like injecting a long square depolarizing current step) and producing 'tonic' like firing in the interneuron. For example, the ~500 ms section trace shown in Figure 3C has stimulation at 30 Hz but only 8 spikes (~16 Hz) and appears to show considerable spike frequency adaptation.

Consequently although the authors describe 'high firing rates' in interneurons we do not actually know what frequency the cells are firing at. The optimum solution would be to reanalyse the data/perform further experiments to determine the actual mean interneuron firing rates obtained by optogenetic stimulation at each frequency and to show this in the manuscript.

These problems somewhat undermine their claims for a frequency-dependent switch from phasic to tonic inhibition (pg 13 ln 7) since it is unclear how the increasing stimulation frequency actually relates to increasing interneuron firing and GABA release.

This also invalidates the interpretation of the data shown in Figure 3H on 'release probability' since at higher frequencies the reduction is most likely due to the discrepancy between the number of light stimulation pulses and the number of spikes the interneuron actually fires rather than a reduction in neurotransmitter release probability following an action potential.

The reviewer makes a number of insightful points that have prompted us to undertake a number of additional experiments that address these “serious concerns”. We now include a direct comparison between interneuron firing rates obtained during optogenetic stimulation and the AP firing rates produced by steady-state current depolarization is now shown in Figure 3 B & C. We believe this illustrates the ability of dLGN-INs to reliably follow the optogenetic stimulation protocols at least up to 30 Hz although there is cell to cell variability as shown in the linear regression analysis. We also agree with the reviewer regarding superimposition of photocurrents but, we do not believe this methodological point necessarily, undermines our main conclusions. At the higher LED stimulation rates superimposition of photocurrents does occur as shown in Figure 3B. However,

optogenetic stimulation at these rates still results in time locked APs in the dLGN-INs and the failure of IPSCs is, we believe, more likely due to transmitter depletion, or postsynaptic receptor desensitization in the relay neuron. However, we now see that the use of the term “release probability” is very misleading, and we have replaced this term with “IPSC response probability” in Figure 3H. Our apologies for the lack of clarity in the previous manuscript.

- 2) The demonstration of frequency dependent modulation of extrasynaptic δ -containing GABA receptors in TC neurons by dLGN interneurons is not novel. This has already been demonstrated by Errington et al., 2011 who showed using pharmacological and electrical stimulation (of retinal inputs to the triad) that activation of mGluRs on interneuron dendrites increases IPSC frequency and tonic current in TC neurons. This study largely confirms these earlier findings using an optogenetic approach. I think it is necessary for the authors to acknowledge this in more detail. Currently the manuscript gives the impression that this is the first demonstration of modulation of tonic GABAergic inhibition in TC neurons by interneurons in the dLGN, which it is not.

We apologise if this impression was given. We agree that Errington et al 2011 demonstrated the recruitment of phasic and tonic inhibition by DHPG in dLGN but absence of this effect in VB. This observation was interpreted in terms of activation of dLGN-INs. We now refer to this work earlier in the modified introduction to make our position absolutely clear. However, the Sox14 mouse enabled us to directly stimulate dLGN-INs. We have also performed new experiments simultaneously recording from Sox 14 dLGN-INs and thalamic relay neurons and this data is presented in the new figure 1. These observations greatly strengthen our conclusions and add to the novelty of this study. We show that increased AP firing rates in a single dLGN-IN increases the tonic conductance in a highly reliable manner. This illustrates a number of important points, not least the fact that the firing of a single dLGN-IN is sufficient to alter the ambient GABA levels experienced by surrounding relay neurons.

- 3) I do not see any evidence presented to support the idea that tonic inhibition mediated by GABA release from interneurons 'will regulate the gain of thalamic relay neurons.' Could the authors please **revise this appropriately to reflect the fact that this is speculation.**

We have now included new experiments that demonstrate the impact of the tonic conductance on thalamic relay neuron excitability (Figure 3I).

Minor points

- 1) To distinguish putative interneurons from thalamocortical neurons in the dLGN it would be useful for the authors to comment on the presence or absence of low threshold spike bursts in these cell types since all TC neurons in dLGN produce LTS bursts whereas interneurons do not. If the authors have this data available I think it would help to support the electrophysiological characterization of the cells.

Furthermore, interneurons in dLGN have slower membrane time constants - do the authors also see longer membrane time constants in GFP+ cells in the Sox14GFP/+ mouse.

In my opinion these are more obvious electrophysiological identifiers (along with input resistance as the authors have shown) for interneurons versus TC cells in dLGN than the I-V relationship or spike firing frequency profile.

This is not a critical point and I am convinced based on the electrophysiological presented that the GFP+ cells are interneurons but I think these suggestions could improve the manuscript.

In response to reviewer 1 we have now included morphological description of the two cell-types. We feel this strengthens this aspect of the work.

2) In Figure 1F both the GFP+ and GFP- cells seems to have quite depolarized resting membrane potentials (injected current 0 pA) for these types of cells in vitro. In the Methods I could not find out if the membrane potentials quoted are corrected for the liquid junction potential. Could the authors please include this in a revised manuscript.

We did not correct for the LJP. We consistently find that the interneuron rest at more depolarised potentials (-50 mV) compared to relay neurons (-60 mV). Indeed, we often observed spontaneous AP firing in the interneurons.

3) The reduction in GABAergic inhibition in TC neurons in the absence of GAD67 expressing dLGN interneurons in Sox14GFP/GFP mice, although not particularly surprising, indicated, at least in brain slices under basal conditions, that interneurons may provide the lion's share of inhibition in dLGN. I think it would be sensible, however, where the authors give the figure of 30% (pg 5, ln 25) of inhibition coming from TRN that they reiterate that this is in the slice and that the in vivo situation could be markedly different.

We agree, and have included a statement related to this fact in the revised manuscript (page 6, line 25-26).

4) Interpretation of the DHPG experiment is complicated. The authors suggest that dLGN interneuron terminals are 'confined within the glomerular arrangement of the thalamic triad' (pg 6 ln 2). It is my understanding that this may be true for dendrodendritic F2 terminals but is not true for conventional axonal F1 terminals. The action of DHPG on dLGN interneurons is mediated by F2 terminals since F1 terminals do not express mGluRs. Consequently the lack of effect of DHPG in Sox14GFP/GFP could be interpreted to stem from a selective disruption of dendritic signalling rather than a complete loss of functional interneurons, especially since some interneurons do remain in the dLGN of Sox14GFP/GFP mice (Figure 2E and pg 4 ln 19). I personally think this is unlikely given the other data presented but I feel some rewriting and acknowledgment of the previous literature here would help.

We agree with this interpretation and have included a modified statement in the revised manuscript to make our position clear (page 7, line 8).

5) Pg 3 ln 13 - change 'that they contain GABA neurotransmitter' to 'they contain the neurotransmitter GABA.'

Done

6) It is not immediately obvious why the authors perform anterograde labelling using injection of Alexa-594 conjugated CTb. I understand it is to identify the visual thalamic nuclei but I think a sentence explaining why they have done it to make it immediately clear to the reader would help.

We have now added a sentence explaining the rationale behind the tracing of the optic tract.

7) Pg 4 ln 23 - the authors say that 'Sox14+ neurons in the dLGN are the only resident inhibitory cell type'. I think this conclusion needs to be toned down slightly given that in Sox14 knockout mice there are still some GFP positive cells in the dLGN and expression of Gad1 and the presence of GABA are 'virtually' but not completely undetectable. Also on pg 7 ln 17-18 the authors say 'virtually all local interneurons'. This is inconsistent with the statement above and the authors should try to be consistent throughout.

Done (see above)

8) Statistics. I could not find any description of the decisions behind why particular statistical tests were used or which analysis software was used to perform the tests in the Results or Methods. I may have missed it but the authors should check and insert a section on statistics into the Methods if appropriate. Also check the figure legends as some did not describe what particular error bars on graphs represented as required by the journal. Finally, ensure consistency in descriptions (e.g. in Figure 2 - mean {plus minus} SEM is used whereas in Figure 5 we see average {plus minus} SE).

This has been corrected

9) The epifluorescence image in Figure 1D is not particularly clear. Could the authors image the cells using a confocal or 2-photon microscope to present a clearer image? Furthermore, if the authors are contrasting the somatodendritic morphology of the GFP+ (putative interneurons) and GFP- (putative TC cells) to aid in cell classification they need to show a comparison fill for both cell types and ideally some quantitative measurement (i.e. soma size/dendritic length).

We have replaced these images with morphological reconstructions to make this distinction.

Finally, I found the description given of the relative morphology of interneurons and TC cells confusing. In both rat and mouse dLGN I would not consider the morphology of interneurons to be compact. They typically have much longer dendrites than TC neurons. The reason for the higher input resistance and lower membrane capacitance is due to the smaller soma size and thinner dendrites (which I think is what the authors mean) but I would not consider the somatodendritic morphology of these interneurons to be compact either physically or electrotonically. I would request that the authors carefully reword this section to clarify this point.

This is exactly what we meant and inclusion of the morphological reconstructions should make this point more clearly in the updated manuscript.

Reviewer #4 (Remarks to the Author)

The Jager et al., study on "Tectal derived Sox14+Otx2+ interneurons contribute to phasic and tonic inhibition in the visual thalamus" is an important and timely contribution to the field. It describes a previously unappreciated population of Sox14+Otx2+ dLGN-INs that control thalamic relay neuron excitability. The paper describes the mechanistic details of how these neurons regulate the gain of thalamic relay neurons through recruitment of extrasynaptic high-affinity GABA_A receptors. The paper provides the first experimental evidence for a tectal origin of thalamic interneurons and redefines dLGN-INs by their expression of Otx2 and Sox14.

The authors describe that most Sox14+ neurons migrate into the dLGN in a dorsal to ventral direction. It would be interesting to know more about the:

A: Clonal relations and exact lineage. Some future clonal analysis should be done (in due course).

B: Examine the possibility that retinal activity regulates this migration to dLGN in enucleation or pharmacological manipulation studies.

The paper raises important questions about the utility of the prosomeric model in the finer connectivity in the diencephalon. Perhaps this is not the place to go into the details of these issues, but I hope the authors shall consider contributing a more detailed review to this specific issue.

I have little to criticize. The paper was put together with care and attention. All figures tell a story and the conclusions are sound. The paper opens up several important developmental and evolutionary/comparative questions and I have no doubt that it will attract general readership.

We are glad that this reviewer is satisfied with the quality of the data and finds our results novel and of sufficient general interest to attract a broad readership.

REVIEWERS' COMMENTS:

Reviewer #1 (Remarks to the Author):

This manuscript is very much improved, and most concerns raised in the initial submission have been carefully addressed by the authors. The new data are clean and compelling, and the paired recording experiments make the point about tonic activation particularly compellingly.

The only issue for the authors to resolve now is the disparity between two findings. The first, with paired recordings, suggest that IN-TC (gpf+→gfp-) connections do not contribute much to synaptic events, as the response is almost completely tonic. Conflicting with this result is the major effect of sox14 deletion on ****synaptic**** responses in figure 2, which indicates that most of the spontaneous synaptic events arise from INs.

I think it is important to address this disparity, at least with a theoretical discussion regarding how it might come about.

Reviewer #2 (Remarks to the Author):

I am happy to comment that the authors have now addressed all the concerns in the original manuscript in a convincing manner. Particularly, the new finding that midbrain-specific deletion of Gata2 results in the loss of dLGN interneurons is a strong evidence for the mesencephalic origin of these cells.

Reviewer #3 (Remarks to the Author):

Review of Revised Version of Jager et al., 2016.

I feel that the authors have made a considerable number of improvements to the manuscript and have in my opinion satisfactorily addressed most of the concerns raised by myself and the other reviewers. I would like to congratulate them for their efforts. I will not comment in further detail on each point and on some issues I feel the other reviewers are more suitably qualified to deal with those points. Overall, considering my previous review I would now be happy for this manuscript to be published in Nature Communications.

However, I still have a few small problems with the interpretation of the electrophysiology and optogenetic experiments despite the number of extra experiments and analyses included by the authors. If the authors could address these issues I feel it would help the final manuscript.

1) Specifically, I would still like to see a simple graph showing the relationship between the blue light pulse stimulation frequency and the actual firing rate of the interneurons. The authors show an example of such an experiment in the revised Figure 3B where the firing rate of a Sox14+ interneurons is able to follow the 30 Hz blue light stimulation pulses but do not give any statistical appreciation of how typical this response is. I am convinced that more blue light stimulation increases the interneuron firing rate and that the rate of firing achieved with light stimulation can match that obtained by current pulse injection (Figure 3C) but it would be really useful to have a quantifiable idea of how the blue light stimulation frequency relates to the firing frequency of the interneurons. Put simply does a 20Hz stimulation produce 20+/-1 Hz firing in interneurons or 10+/-1 Hz firing? This is important in interpreting the reduction in 'IPSC response probability'.

2) The weighted decay time constants of the IPSCs in GFP- neurons (TC neurons) produced by blue light stimulation of GFP+ neurons (interneurons) is around 20-25 ms (Figure 3F). This means the IPSCs take considerably more than 50 ms to decay to baseline. At 20 Hz the inter-stimulus interval is 50 ms. I wonder how much of the increased tonic conductance reported is due to simple summation of IPSCs? Could the authors comment on this in the manuscript?

3) The experiments in Figure 2F and G appear to suggest quite strongly that Sox14 GFP+ interneurons can modulate tonic inhibition at high firing rates in neighbouring TC neurons to which they are not directly connected either through F1 (axonic) or F2 (dendrodendritic) synapses since increased firing in the presynaptic cell is not reflected by increased IPSCs in the postsynaptic cell. In Figure 3 tonic conductance is increased in TC neurons where they do receive direct synaptic input from interneurons (as evidenced by stimulus dependent IPSCs). This suggests that tonic inhibition mediated by interneurons in the dLGN is more like 'volume transmission' rather than necessarily requiring specific cell to cell connectivity. Could the authors comment on whether tonic conductance is more strongly modulated if the interneuron is directly connected to the postsynaptic TC neuron? Does this mean that tonic inhibition in the dLGN has a 'direct' and 'indirect' component? It might be nice to see some more discussion dedicated to these point and the potential consequences for the visual thalamus. However, I would leave this to the discretion of the authors.

4) Minor point: were the patch clamp experiments done at 35 degs or room temperature? I couldn't seem to find this in the text.

Response to reviewers' comments:

Reviewer #1 (Remarks to the Author):

This manuscript is very much improved, and most concerns raised in the initial submission have been carefully addressed by the authors. The new data are clean and compelling, and the paired recording experiments make the point about tonic activation particularly compellingly.

The only issue for the authors to resolve now is the disparity between two findings. The first, with paired recordings, suggest that IN-TC (gpf+→gfp-) connections do not contribute much to synaptic events, as the response is almost completely tonic. Conflicting with this result is the major effect of sox14 deletion on ****synaptic**** responses in figure 2, which indicates that most of the spontaneous synaptic events arise from INs.

I think it is important to address this disparity, at least with a theoretical discussion regarding how it might come about.

We would like to thank **Reviewer #1** for their positive comments on the revised submission. The reviewer requested we explain a possible disparity between absence of stimulus evoked synaptic events in the paired recording data and the reduced frequency of spontaneous synaptic events following sox14 deletion. We agree these observations do require some explanation and we believe there are two possible reasons for this apparent contradiction. Firstly, spontaneous synaptic events are likely to reflect the vesicular GABA release from many dLGN-INs that connect onto a single thalamic relay neuron whereas the paired recordings examine the contribution of a single dLGN-IN. It is, therefore, not surprising that paired recordings fail to reveal the low number of connections made by an individual dLGN-INs. Secondly, evoking release at high frequencies from a single dLGN-IN is likely to result in considerable synaptic fatigue reflecting presynaptic vesicular depletion and postsynaptic desensitization that will reduce the likelihood of observing a synaptic response in the paired recording data. **These theoretical considerations have now been included in the revised manuscript (see page 12, lines 10-14).**

Reviewer #3 (Remarks to the Author):

Review of Revised Version of Jager et al., 2016.

I feel that the authors have made a considerable number of improvements to the manuscript and have in my opinion satisfactorily addressed most of the concerns raised by myself and the other reviewers. I would like to congratulate them for their efforts. I will not comment in further detail on each point and on some issues I feel the other reviewers are more suitably qualified to deal with those points. Overall, considering my previous review I would now be happy for this manuscript to be published in Nature Communications.

However, I still have a few small problems with the interpretation of the electrophysiology and optogenetic experiments despite the number of extra experiments and analyses included by the authors. If the authors could address these issues I feel it would help the final manuscript.

1) Specifically, I would still like to see a simple graph showing the relationship between the blue light pulse stimulation frequency and the actual firing rate of the interneurons. The authors show an example of such an experiment in the revised Figure 3B where the firing rate of a Sox14+ interneurons is able to follow the 30 Hz blue light stimulation pulses but do not give any statistical appreciation of how typical this response is. I am convinced that more blue light stimulation increases the interneuron firing rate and that the rate of firing achieved with light stimulation can match that obtained by current pulse injection (Figure 3C) but it would be really useful to have a quantifiable idea of how the blue light stimulation frequency relates to the firing frequency of the interneurons.

Put simply does a 20Hz stimulation produce 20+/-1 Hz firing in interneurons or 10+/-1 Hz firing? This is important in interpreting the reduction in 'IPSC response probability'.

2) The weighted decay time constants of the IPSCs in GFP- neurons (TC neurons) produced by blue light stimulation of GFP+ neurons (interneurons) is around 20-25 ms (Figure 3F). This means the IPSCs take considerably more than 50 ms to decay to baseline. At 20 Hz the inter-stimulus interval is 50 ms. I wonder how much of the increased tonic conductance reported is due to simple summation of IPSCs? Could the authors comment on this in the manuscript?

3) The experiments in Figure 2F and G appear to suggest quite strongly that Sox14 GFP+ interneurons can modulate tonic inhibition at high firing rates in neighbouring TC neurons to which they are not directly connected either through F1 (axonic) or F2 (dendrodendritic) synapses since increased firing in the presynaptic cell is not reflected by increased IPSCs in the postsynaptic cell. In Figure 3 tonic conductance is increased in TC neurons where they do receive direct synaptic input from interneurons (as evidenced by stimulus dependent IPSCs). This suggests that tonic inhibition mediated by interneurons in the dLGN is more like 'volume transmission' rather than necessarily requiring specific cell to cell connectivity. Could the authors comment on whether tonic conductance is more strongly modulated if the interneuron is directly connected to the postsynaptic TC neuron? Does this mean that tonic inhibition in the dLGN has a 'direct' and 'indirect' component? It might be nice to see some more discussion dedicated to these point and the potential consequences for the visual thalamus. However, I would leave this to the discretion of the authors.

4) Minor point: were the patch clamp experiments done at 35 degs or room temperature? I couldn't seem to find this in the text.

Reviewer #3 commented on the considerable improvements made to the manuscript and we would like to thank the reviewer for their contribution to this process. The reviewer requested a simple

graph showing the relationship between the LED stimulation frequency and the firing rate of the dLGN-INs to simplify interpretation of the reduction in 'IPSC response probability'. This plot has been included in a revised version of Figure 3. We have also included a further reference to this analysis in the revised text (page 5, line 18-23) to stress how the ability of a dLGN-IN to follow the LED flashes is associated with the maximum AP frequency observed during current injection. Therefore, those cells that cannot follow LED frequencies above 10 Hz also exhibit lower firing rates in response to current injection. We agree with the reviewer that summation of IPSCs will contribute to the tonic conductance generated at high LED stimulation rates. However, it is clear that in the absence of LED stimulation the tonic conductance is still sizeable. This can be demonstrated using the data illustrated in panel 3E where the LED stimulation rates are only 0.1 Hz. A steady-state tonic current is clearly apparent throughout the current record both before and after the evoked IPSC as the steady-state inward current is reduced by gabazine. A statement designed to clarify this point has been included in the revised figure legend. We also agree with the reviewer that the tonic inhibition mediated by dLGN-INs is more like 'volume transmission' and does not necessarily require specific cell to cell connectivity. However, we cannot at present comment on whether tonic conductance is more strongly modulated if the interneuron is directly connected to the postsynaptic TC neuron. Therefore, we would like to leave discussion of this important point for a future study. Finally, we have included a statement on recording temperature in the revised methods section (page 16, line 30).